# Young domestic chicks spontaneously represent the absence of objects

Eszter Szabó[1]*, Cinzia Chiandetti[2], Ernő Téglás[1], Elisabetta Versace[3], Gergely Csibra[1,4], Ágnes Melinda Kovács[1], Giorgio Vallortigara[5]

[1]Department of Cognitive Science, Central European University, Vienna, Austria; [2]Department of Life Sciences, University of Trieste, Trieste, Italy; [3]School of Biological and Behavioural Sciences, Department of Biological and Experimental Psychology, Queen Mary University of London, London, United Kingdom; [4]Department of Psychological Sciences, Birkbeck, University of London, London, United Kingdom; [5]Center for Mind/Brain Sciences, University of Trento, Rovereto, Italy

**Abstract** Absence is a notion that is usually captured by language-related concepts like zero or negation. Whether nonlinguistic creatures encode similar thoughts is an open question, as everyday behavior marked by absence (of food, of social partners) can be explained solely by expecting presence somewhere else. We investigated 8-day-old chicks' looking behavior in response to events violating expectations about the presence or absence of an object. We found different behavioral responses to violations of presence and absence, suggesting distinct underlying mechanisms. Importantly, chicks displayed an avian signature of novelty detection to violations of absence, namely a sex-dependent left-eye-bias. Follow-up experiments excluded accounts that would explain this bias by perceptual mismatch or by representing the object at different locations. These results suggest that the ability to spontaneously form representations about the absence of objects likely belongs to the initial cognitive repertoire of vertebrate species.

*For correspondence: szaboe@ceu.edu

**Competing interest:** The authors declare that no competing interests exist.

## Editor's evaluation

The research detailed in this manuscript investigates whether young chicks represent the absence of objects. This work is important to multiple fields of inquiry, such as ethology and neuroscience, and is the first time that this ability has been demonstrated to be exhibited spontaneously, as opposed to after many trials of experience.

## Introduction

Imagine looking at a domino that has four dots on one end and no dots on the other. The ways we can represent specific items (e.g., four dots) have been intensively investigated for decades. First, one can think of these dots as individual objects. Investigations targeting object cognition revealed that roughly four objects can be tracked and maintained in mind simultaneously, even when they are moving or are occasionally occluded (*Kahneman et al., 1992*; *Scholl and Pylyshyn, 1999*). Another way to look at the dots is to encode them as a set of objects. Such encoding is performed by the approximate number system, which provides imprecise representations of sets to pre- and nonlinguistic creatures as well. However, in contrast to the object tracking system, information in this number system is only an approximation of the size of the set and it is sensitive to proportional rather than to absolute differences (*Dehaene, 1997*). Both systems emerge early in the individual development (*Piazza, 2010*) and are shared by several species (e.g., mosquito fish, domestic chicks, rhesus monkeys, and great apes; *Haun et al., 2011*; *Brannon and Merritt, 2011*; *Vallortigara, 2012*; *Brannon and Roitman, 2003*). A

third way to think about the four dots on the domino is as a symbolic (and precise) representation of the number '4' (*Dehaene, 1997*; *Carey, 2009*). Interpreting such symbols clearly requires processing number concepts and being familiar with the specific notation system (e.g., the conventions of dominos or the Arabic numerals) (*Carey, 2009*). Now, let us focus on the other end of the domino. How will the blank square turn into *zero* in our mind? Such a representation may be outside the scope of the abovementioned cognitive systems: 'no object' is not tracked by the visual system, 'no dots' is not proportional to anything, and 'empty space' can denote a number only in special circumstances. Indeed, understanding the absence of something as 'nothing' is frequently related to complex and human-specific concepts, such as zero or linguistic negation. In this work, we focus on the representation of absence that should, however, rely on a more basic capacity and possibly be part of the initial cognitive repertoire of different species and we will target such abilities in 8-day-old domestic chicks.

Clear evidence regarding when human children start representing the absence of objects comes from language development research. Negation conveying absence (e.g., 'all gone') emerges among the first linguistic expressions between 1 and 2 years of life (*Bloom, 1970*; *Pea, 1980*; *Choi, 1988*). Some years later, preschoolers can flexibly use sentential negation to express the absence of something in a numerical context (*Bialystok and Codd, 2000*), and can recruit complex numerical concepts, such as zero (*Wellman and Miller, 1986*; *Merritt and Brannon, 2013*). While by the age of 5 children seem to successfully operate with counterintuitive concepts like zero and nothing, the cognitive foundations of this human capacity are frequently suggested to be grounded in linguistic abilities. How would pre- and nonlinguistic creatures see the zero end of the domino?

Nonhuman animals were found to accommodate stimuli defined by the lack of a stimulant in two main types of tasks: numerical and perceptual decision tasks. Studies involving numerical tasks indicate that monkeys can integrate empty sets with other sets relying on the approximate number system (*Biro and Matsuzawa, 2001*; *Merritt et al., 2009*; *Howard et al., 2018*). For instance, comparisons including empty sets are also subject to distance effects, characteristic to the approximate number system (the smaller the distance between two numerosities, the more errors the subjects make) *Merritt et al., 2009*. Empty sets, similar to other numerosities, are represented in the number specific areas of the monkey brain (*Macaca fuscata*: *Okuyama et al., 2015*; *Macaca mulatta*: *Ramirez-Cardenas et al., 2016*), which provides further evidence for the involvement of this system. Furthermore, an interesting finding suggests that Ai, the chimpanzee learned to use a symbol for zero (*Biro and Matsuzawa, 2001*). However, Ai's performance likely reflected a rather limited conceptual understanding of zero, as she did not show transfer effects when switching from cardinality judgments to ordering tasks.

Although these findings are very impressive, it is still unclear how the approximate number system could represent *exactly no objects*, given that it is specialized for approximating numerosity. Evidence pointing to the possibility that this system may not be appropriate for such encoding comes from studies with preschoolers (*Merritt and Brannon, 2013*) and monkeys *Merritt et al., 2009* who tend to fail to discriminate an empty set from one item. Thus, the question emerges how the representation of *exactly no objects* might be encoded. 'Nothing' is an amount of less than one, but crucially, it can also be thought of as one side of the binary information of presence and absence. The role of the approximate number system in the first, continuous conceptualization of 'nothing' seems unequivocal, however, the binary coding of 'nothing' is a more peculiar subject of investigation. Things can be present or absent, yet how these intuitively simple opposing categories are formed and encoded is largely unexplored. *Bermúdez, 2003* proposed that contrary concepts, like absence/presence might be available even for nonhuman animals. Such contrary concepts encompass alternatives that are mutually exclusive (e.g., nothing can be present and absent at the same time) and may support specific inferences. The availability of contrary concepts in pre- and nonlinguistic animals has not been targeted by researchers, nevertheless, extensive research cumulating for over a century suggests that various species are able to exploit the presence and absence of stimulus (*Pearce, 2011*). However, clear evidence that absence is explicitly represented is scarce. Importantly, not representing a stimulus is not equivalent with representing its absence, in a way that this would be distinguishable from a nonspecific default activation of a system (*de Lafuente and Romo, 2005*; *Merten and Nieder, 2012*). A recent study has targeted this issue, by investigating prefrontal neural activations in monkeys while performing abstract detection decisions regarding the presence and absence of stimuli (*Merten and Nieder, 2012*). Notably, in this study the stimulus presentation phase was separated from a later phase preceding decision. While presence-specific neurons were found to be active when the animal

perceived the stimulus and also later when making a decision, absence-specific neurons showed activation only when the subject decided about absence. This finding, besides providing evidence for forming some representations of absence in monkeys, points to the possibility that different processes are involved in encoding the presence and absence of a stimulus. Asymmetries between performance relying on representing the presence and absence of stimuli were documented in behavioral tasks as well. Pigeons (*Hearst, 1984*; as well as human adults, *Newman et al., 1980*) display feature-positive biases in learning tasks. Pigeons learned relatively easily the relation between the presence of a stimulus and food, but they had difficulties with discovering a similar relation between the absence of a stimulus and food. In line with such asymmetries, human infants automatically detect and keep in mind the presence of objects after occlusion, while they seem to fail to do so with the absence of objects (*Wynn and Chiang, 2016*; *Kaufman et al., 2003*). While in these tasks the representation of an object being present can be supported by the object tracking system (*Kahneman et al., 1992*), it is unclear how a specific object that is absent could be encoded by the same system (note that simply discarding the object file results in no representation whatsoever and it is not equivalent to representing, for instance, 'the lion is absent'). In fact, the representation of absence might be beyond the scope of this system, as it operates with spatiotemporal information of the items, which absent objects do not have. Thus, up to date it is unclear under which circumstances individuals form representations of 'no object', whether such representations can be used for further processing as readily as the presence of a stimulus, and most importantly, whether they can be encoded spontaneously.

Absence is trivial in experience, but peculiar in information processing. While some nonhuman species show success in dealing with the absence of stimuli in experimental tasks involving training or hundreds of trials (*Merten and Nieder, 2012*), it is not yet known whether nonlinguistic creatures can spontaneously rely on such information and what inferences they can draw from it. A possible way to investigate the emergence and the nature of the representation of absence is to target developmentally precocious animals. We addressed these questions by studying naive domestic chicks — creatures that start to search for food soon after hatching, and could make good use of information regarding the presence or absence of potential food sources and social partners.

In four experiments, 8-day-old chicks were placed inside a confining cylinder that had a small circular opening to provide the opportunity of putting through the head and attend the events in the testing arena. The age was determined by the specific paradigm we used in the present work (see the details of imprinting and familiarization with the apparatus in the Materials and methods) and the dependent measures we targeted (looking time and lateralization index). This is also the age when chicks were found to show strong lateralized responses to familiar and unfamiliar objects (*Dharmaretnam and Andrew, 1994*). Chicks were presented with events in which the target object they were imprinted to (for the details see Materials and methods) was either hidden behind a screen or was removed from the arena. Afterwards, the screen was dropped and it revealed an expected outcome (congruent with the previous event, e.g., the object appeared from behind the screen after it was hidden behind the screen) or an unexpected outcome (contradicting the previous events, e.g., the object appeared from behind the screen after it was removed from the arena). If in this latter case chicks represented the object as being absent, and they see an incongruent outcome (the object 'magically' appears), they should show a different behavior compared to when they see the scene-congruent outcome (the object was present and appears expectedly). We measured how long the chicks looked at these outcomes and which eye they used to inspect the scene. Regarding our first measurement, based on former research with human infants (*Baillargeon et al., 1985*) and a study involving adult rooks (*Bird and Emery, 2010*), we expected longer looking for unexpected outcomes. For instance, *Bird and Emery, 2010* have found that rooks looked longer to unexpected events that violate the laws of physics (e.g., objects remaining in the air without any support) compared to expected events (e.g., objects in a support relation with other objects), indicating a violation of their expectation or surprise, a method commonly used in infancy research to study a wide range of competencies.

For our second measurement, we coded which eye the chicks used to inspect the expected and unexpected outcomes. Earlier research suggests that eye usage is modulated by the novelty of the object attended, and also by sex (*Rogers and Anson, 1979*; *Dharmaretnam and Andrew, 1994*; *Vallortigara and Andrew, 1991*). A preferential use of the left eye (mainly feeding the right brain structures) is associated with response to novelty in birds with laterally placed eyes such as domestic chicks (*Rogers et al., 2013*). Note that the selective involvement of structures in the right hemisphere

when attending to novel stimuli is widely documented among vertebrates (review in *Rogers et al., 2013*), being likely a general feature inherited by early chordates (*MacNeilage et al., 2009*). In animals with laterally placed eyes and lack of callosum, such as birds, fish, reptiles, and amphibians the brain asymmetry can be easily documented without any invasive procedure by simply measuring preferences in eye use (*Vallortigara, 2000*; *Vallortigara and Versace, 2017*; *Vallortigara and Rogers, 2020*). In the present study, we expected a left-eye bias to the novel unexpected outcomes compared to the expected ones.

In addition, interestingly, lateralized sex differences have been repeatedly observed in response to novel objects. For instance, *Vallortigara and Andrew, 1991* documented stronger left-eye-mediated choices of unfamiliar stimuli for males compared to females, and different preferences for unfamiliar and familiar objects between sexes. Lateralized sex differences have been found also by *Dharmaretnam and Andrew, 1994*, where unfamiliar stimuli evoked left-eye bias in females, but not in males. *Vallortigara and Andrew, 1991* observed that left-eye (and binocular) males preferred unfamiliar objects, while left-eye (and binocular) females preferred familiar objects, whereas both males and females tested with the right eye did not exhibit significant preferences for familiar or novel stimuli. Hence, a potential modulation of sex in eye use must be considered for the exploration of unexpected vs. expected scenes in our study as well.

## Study 1

In Study 1, we investigated whether the chicks encoded the presence (Experiment 1) and the absence (Experiment 2) of the object behind the screen. In Experiment 1 (Encoding Presence), chicks (n = 27, 13 females, 14 males) watched as the object moved behind the screen till full occlusion, and then they either saw it moving out of the scene (Expected Disappearance condition) or did not see it moving out (Unexpected Disappearance condition). Both scenes ended identically, by the screen falling and revealing an outcome with no object being present (*Figure 1A*). In Experiment 2 (Encoding Absence), the test trials started with the screen in a lowered position. A different group of chicks (n = 28, 15 females, 13 males) observed an object moving to the area behind the lowered screen (Expected Appearance condition) or moving out of the scene (Unexpected Appearance condition) before the screen was raised (*Figure 1B*). Both scenes ended by the screen falling and revealing an outcome with the object being present. We used a repeated measures design; thus, each chick was presented with both the Expected and the Unexpected conditions. We coded the chicks' overall Looking Times and Lateralization Index (the difference between left- and right-eye usage proportional to the total eye usage) in the outcome phases.

## Results

Chicks' overall Looking Times were differently modulated as a function of the outcomes violating or confirming the chicks' expectations about the presence and the absence of the object in Experiments 1 and 2 (*Figure 2A*). We run a repeated measures analysis of variance (ANOVA) on the square root transformed data set. A 2 × 2 × 2 repeated measures ANOVA with Experiment (Experiment 1: Encoding Presence vs. Experiment 2: Encoding Absence), Outcome (Expected vs. Unexpected), and Sex (Female vs. Male) as factors yielded a significant interaction between Experiment and Outcome ($F_{1, 51}$ = 4.244, p = 0.045, $\eta_p^2$ = 0.077) with no other effects (IBM SPSS Statistics 20). In Experiment 1, chicks' looking behavior seemed to be in line with the prediction of longer looking times for unexpected outcomes (untransformed means: Unexpected Disappearance: $M$ = 15.421; SD = 7.34; Expected Disappearance: $M$ = 12.671, SD = 6.566; Scheffé test p = 0.078, $\eta^2$ = 0.037); however, this was not the case for Experiment 2 (Unexpected Appearance: $M$ = 12.906; SD = 8.409; Expected Appearance: $M$ = 14.556, SD = 9.351; Scheffé test, p = 0.273, $\eta^2$ = 0.008). Subjects looked longer to the Unexpected outcomes in Experiment 1 compared to Experiment 2 (Scheffé test, p = 0.028, $d$ = 0.318). Human infants show similar looking patterns to such scenes (i.e., longer looks to violations of presence but not to that of absence; *Wynn and Chiang, 2016*; *Kaufman et al., 2003*), and it was suggested that they are more sensitive to violations of presence compared to violations regarding the absence of objects.

A 2 × 2 × 2 repeated measures ANOVA performed on the Lateralization Index revealed a significant Experiment by Outcome interaction ($F_{1, 51}$ = 4.652, p = 0.036, $\eta_p^2$ = 0.084), with no other significant effects. Pairwise comparisons revealed that in the Unexpected Outcome condition subjects displayed

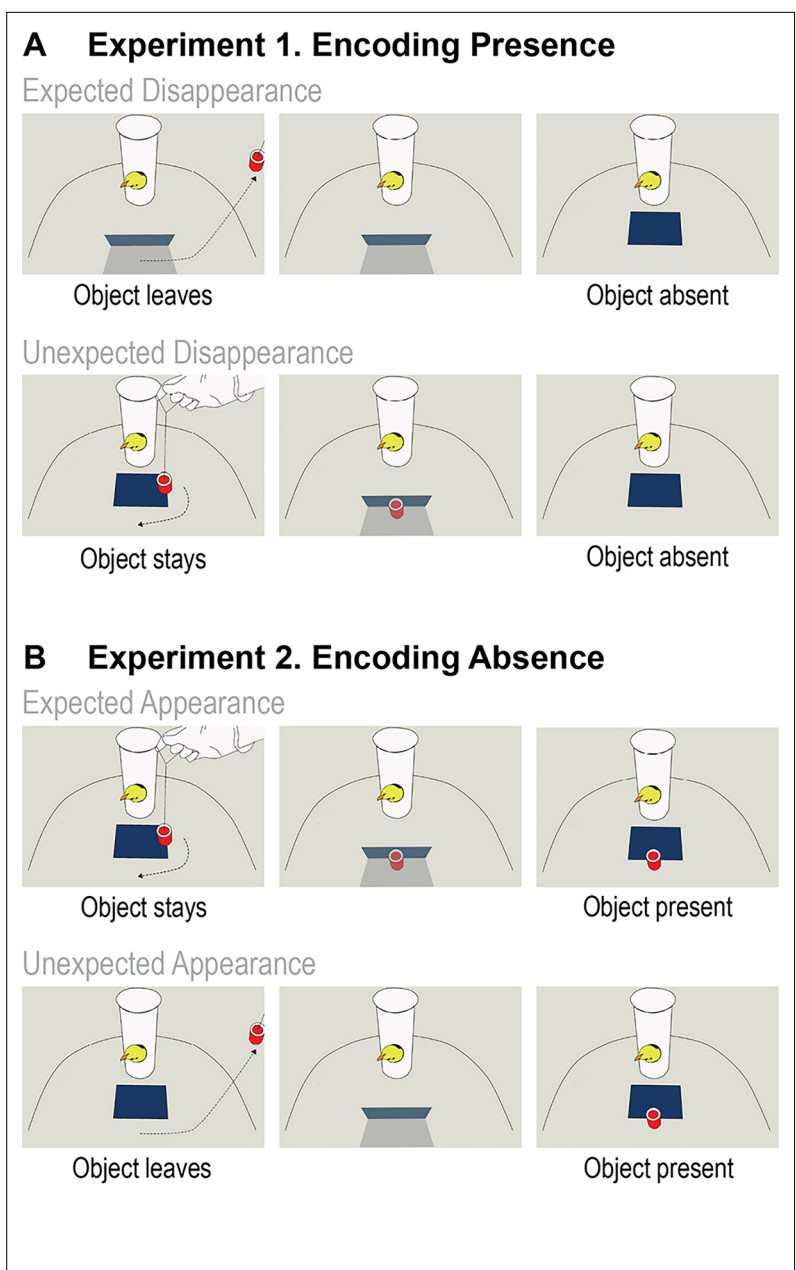

**Figure 1.** Schematic illustration of the events in Experiments 1 and 2. (**A**) Experiment 1 – Encoding Presence. The upper panels depict the events in the Expected Disappearance condition, where the target object was removed from the arena before the screen was lowered revealing the empty space behind it. The lower panels depict the events in the Unexpected Disappearance condition, where the target object was placed behind the screen visibly to the chick but then it was secretly removed from the arena. When the screen was lowered, it revealed the empty space behind. (**B**) Experiment 2 – Encoding Absence. The upper panels depict the events in the Expected Appearance condition, where the target object moved behind the screen and when the screen was lowered, it revealed the presence of the object. The lower panels depict the Unexpected Appearance condition, in which the target object was visibly removed from the arena, and then the vertical position of the screen was restored. Afterwards, the target object was secretly reintroduced into the arena, and when the screen was lowered, it revealed the target object.

a greater left-eye bias in Experiment 2 (Encoding Absence, $M = 0.193$, SD = 0.517) compared to Experiment 1 (Encoding Presence, $M = -0.138$, SD = 0.459; Scheffé test, p = 0.023, $d = 0.677$). The other pairwise comparisons did not reach significance, though the pattern of eye usage was congruent with our predictions in Experiment 2 (greater left-eye bias for the unexpected outcomes:

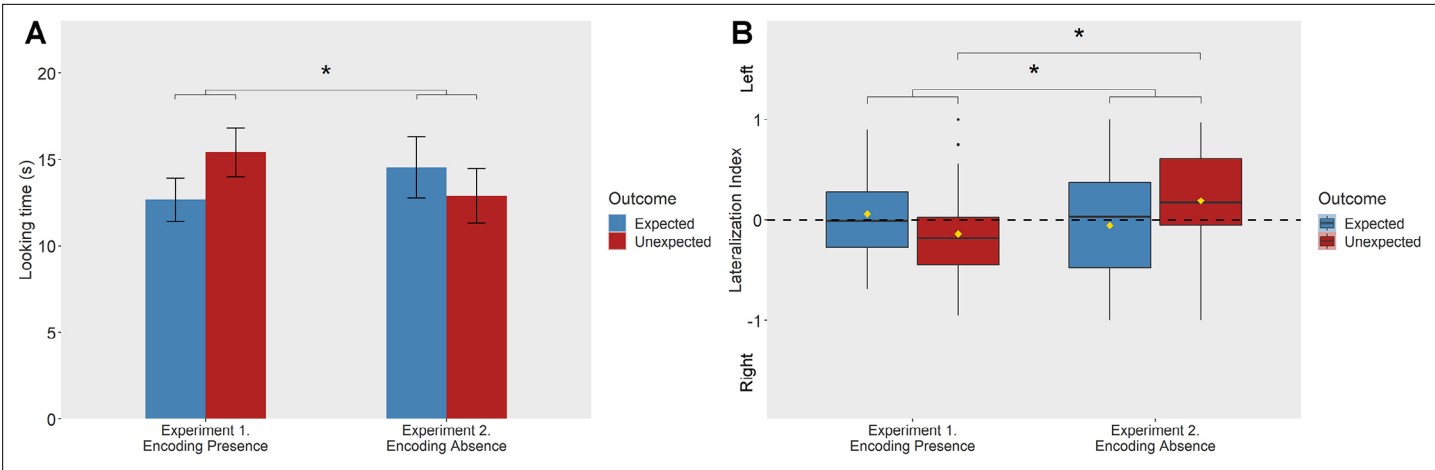

**Figure 2.** Results of Experiments 1 and 2. (**A**) Mean Looking Times elicited by Expected and Unexpected outcomes in the two experiments. The asterisk indicates a significant interaction between Outcome (Expected/Unexpected) and Experiment ($F_{1,51}$ = 4.244, p = 0.045) and a significantly higher looking time observed for the Unexpected outcome in Experiment 1 (Encoding Absence) than in Experiment 2 (Encoding Presence) (Scheffé test, p = 0.028). Error bars represent standard error of the mean. (**B**) Lateralization Index as a function of Experiment and types of outcome. Asterisks indicate a significant interaction between Outcome (Expected/Unexpected) and Experiment ($F_{1,51}$ = 4.652, p = 0.036) and a significantly higher left-eye bias observed for the Unexpected outcome in Experiment 2 (Encoding Absence) than in Experiment 1 (Encoding Presence) (Scheffé test, p = 0.023), suggesting that the Lateralization Index is sensitive to violations of expectation regarding the absence of objects. For box plots, the horizontal line represents the median, yellow diamonds depict the mean values, box height depicts first and third quartiles, and vertical lines represent the 95th percentile. Dots represent the outliers of the data set.

Unexpected Appearance, *M* = 0.193, SD = 0.518; Expected Appearance *M* = −0.054, SD = 0.518; Scheffé test, p = 0.091, $\eta^2$ = 0.05), but not in Experiment 1.

These data suggest that the Lateralization Index may be sensitive to violations of expectation regarding the absence of objects, but not regarding the presence of objects (*Figure 2B*). A positive Lateralization Index reflects a left-eye bias, which was previously linked to detecting novel outcomes (*Rogers and Anson, 1979*; *Dharmaretnam and Andrew, 1994*), however in our case it seems to be specific to violations of expectations about the object's absence. Thus, while there is currently no report suggesting that human infants (*Wynn and Chiang, 2016*; *Kaufman et al., 2003*) or other animals would spontaneously encode absence, the Lateralization Index in the present study points to 8-day-old chicks' ability to encode and form expectations about 'no objects' at a particular location. The left-eye bias, found in response to the unexpected appearance of the object, likely reflected the chicks' exploration, and attempt of identification of the unexpectedly emerging 'new' item. Importantly, this item was new only if chicks had encoded that the identically looking familiar item left, and thus it was absent from the scene. They likely investigated the 'new' object more carefully with their left eye because they expected the old object to be absent.

Interestingly, in contrast to the left-eye bias, overall looking time did not seem to be sensitive to the unexpected appearance of the object, and therefore to absence. We assumed that if chicks are sensitive to both presence and absence violations, they should show surprise (and produce longer looking times) in the Unexpected outcomes compared to the Expected outcomes in both experiments. Instead, we found an Experiment by Outcome interaction. Only Experiment 1 (Encoding Presence) revealed a looking time that patterned with our prediction, but not Experiment 2 (Encoding Absence). The present work was, however, to our knowledge, the first attempt to measure young chicks' looking time in violation of expectation paradigms, thus it may be difficult to interpret this pattern. It may be the case that in our setting overall looking times did not reveal clear differences due to statistical variability.

To sum up, in Study 1 we found asymmetric patterns of looking times and lateralization in response to outcomes congruent or violating chicks' expectations of presence and absence. Furthermore, we found that the left-eye bias seems to reflect chicks' ability to form an expectation about the absence of an object. In Study 2, we targeted the cognitive processes that could support the representation of absence.

## Study 2

In two additional experiments, we focused on the left-eye bias effect observed in Experiment 2. We aimed on the one hand to strengthen our findings regarding chicks' absence encoding, and on the other hand to test the possible representational processes underlying such a behavior. According to one possibility, chicks in Experiment 2 (Encoding Absence) might have not formed an actual representation that there was no object behind the screen, instead their reaction to the unexpected appearance could have been perceptually driven and derived from detecting the mismatch between the memory of the empty space behind the screen (i.e., an iconic, picture-like representation of the empty floor, and the wall of the testing arena) and the perceived outcome (i.e., the scene with the object). This possibility, however, would predict no left-eye bias in a situation where there was no possibility for perceptually encoding the empty space. In contrast, left-eye bias without relying on the percept of an empty space would point to a more complex mental representation of absence that is inferred from the sequence of events (object going in/going out) (Experiment 3. Absence vs. Perceptual Comparison). A second alternative explanation might be that the chicks did not encode the absence of the object, but tracked the location of the target object even when it left the scene, and encoded its presence somewhere outside the arena, and the left-eye bias reflected their 'surprise' of seeing this object at an unexpected location (i.e., behind the screen, inside the arena). According to this alternative, an outcome which would feature a different object behind the occluder should not elicit a left-eye bias (Experiment 4. Absence vs. Tracking). In the second study, Experiments 3 and 4 test these alternative explanations, respectively, and aimed at replicating the findings from Experiment 2.

Experiment 3 (Absence vs. Perceptual Comparison) followed the procedure of Experiment 2. However, unlike in Experiment 2, each trial started with the screen in upright position preventing the chicks ($n$ = 31, 15 females, 16 males) to see the space occluded by the screen at the beginning of the trial (*Figure 3A*). This manipulation aimed at testing chicks' ability to update their expectation about what is (not) behind the screen without giving them the opportunity to perform perceptual comparison between an initial empty scene and the outcome. Without clear perceptual evidence, we expect chicks to arrive at encoding absence by first representing the object being behind the screen, and then inferring the outcome (i.e., absence) after the object being removed from behind the screen. Based on the observation of these events chicks may compute the absence objects and show similar behavior (i.e., left-eye bias in response to the unexpected appearance of the object) as we observed in Experiment 2.

Experiment 4 (Absence vs. Tracking) ($n$ = 22, 9 females, 13 males) was the same as Experiment 3, except that in the Unexpected Appearance condition the target object that moved behind the screen and then left the scene was different from the object that appeared when the screen was lowered in the outcome phase (*Figure 3B*). Finding a second object at the previously empty location should not be surprising if chicks simply track the location of the first target object. However, finding an object at a location that is represented as empty would lead to surprise even if this object is different from the one that has left. Thus a left-eye bias in the Unexpected Appearance condition of Experiment 4 would provide evidence for chicks' capacity to form expectation about absence of objects at a specific location that must be therefore empty. In contrast, the lack of such response would rather point to object tracking processes underlying chicks' left-eye bias in Experiment 2 that resulted in encoding the presence of the first object at a different location (outside the scene).

## Results

We analyzed the Lateralization Index of Experiments 3 and 4 with a 2 × 2 × 2 repeated measures ANOVA with Experiment, Outcome, and Sex as factors. There was a main effect of Outcome ($F_{1, 49}$ = 4.29, p = 0.044, $\eta_p^2$ = 0.081), revealing more positive values (more usage of the left eye) in response to unexpected outcomes ($M$ = 0.087, SD = 0.5) compared to expected outcomes ($M$ = −0.045, SD = 0.459), similar to Experiment 2. Additionally, a significant interaction was observed between Outcome and Sex ($F_{1, 49}$ = 4.804, p = 0.033, $\eta_p^2$ = 0.089), indicating females' sensitivity to events violating their expectations regarding absence of entities (Unexpected Appearance, $M$ = 0.189; SD = 0.5; Expected Appearance, $M$ = −0.133, SD = 0.5; post hoc Scheffé test, p = 0.007, $\eta^2$ = 0.095), while males showed no such difference (Unexpected Appearance, $M$ = −0.01; SD = 0.49; Expected Appearance, $M$ = −0.005, SD = 0.377; post hoc Scheffé test, p = 0.918, $\eta^2$ = 0.000).

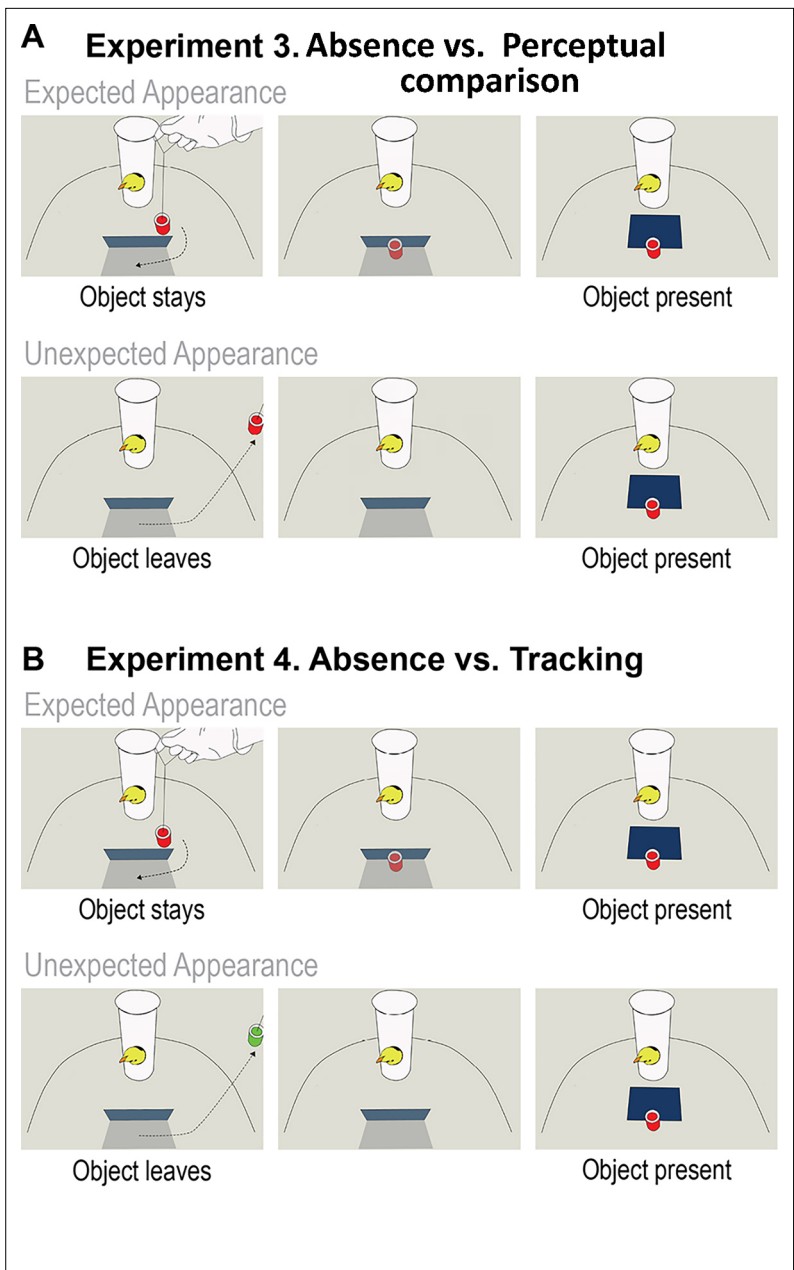

**Figure 3.** Schematic illustration of the events in Experiments 3 and 4. (**A**) Experiment 3 – Absence vs. Perceptual Comparison. The upper panels depict the event in the Expected Appearance condition, where the target object moved behind the screen and when the screen was lowered, it revealed the object behind. The lower panels depict the events in the Unexpected Appearance condition, in which the object first moved behind the screen and then it was visibly removed from the arena. Afterwards, the object was secretly placed behind the screen and when the screen was lowered, it revealed the object. Note that, in both conditions, the screen's initial position was vertical. (**B**) Experiment 4 – Tracking vs. Absence. The upper panels depict the events in the Expected Appearance condition, which was identical to the same condition of Experiment 3. The lower panels depict the Unexpected Appearance condition where the (green) target object first moved behind the screen and then it was visibly removed from the arena. Afterwards, the red object was secretly reintroduced behind the screen. In the outcome phase, the screen was lowered and the red object appeared.

Next, we included in a 3 × 2 × 2 repeated measures ANOVA the three Experiments that targeted absence encoding (Experiments 2–4), evaluating the Lateralization Index with Experiment, Outcome, and Sex as factors (*Figure 4*). We found a main effect of Outcome ($F_{1, 75}$ = 6.273, p = 0.014, $\eta_p^2$ = 0.077), reflecting more positive values (more usage of the left eye) for the Unexpected Appearance

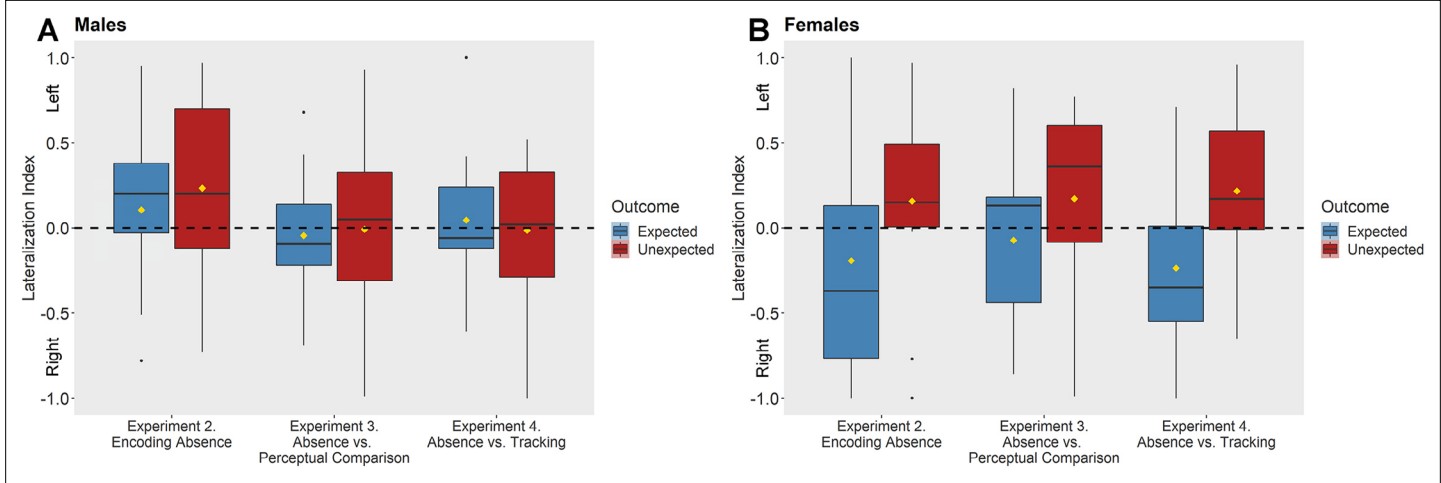

**Figure 4.** Results of Experiments 2–4. (**A**) Mean Lateralization Index elicited by Expected and Unexpected outcomes for male chicks. (**B**) Mean Lateralization Index elicited by Expected and Unexpected outcomes for female chicks. An 3 × 2 × 2 analysis of variance (ANOVA) revealed a main effect of Outcome ($F_{1, 75}$ = 6.273, p = 0.014), reflecting higher level of left-eye bias for the Unexpected compared to the Expected outcome and a significant interaction between Outcome and Sex ($F_{1, 75}$ = 4.157, p = 0.045) indicating females' more pronounced differentiations between the outcomes. The horizontal line is the median, yellow diamonds depict the mean values, box height depicts first and third quartiles, and vertical lines represent the 95th percentile. Dots represent the outliers of the data set.

outcomes ($M$ = 0.123, SD = 0.506) compared to the Expected Appearance ($M$ = −0.048, SD = 0.498) outcomes. There was also a significant interaction between Outcome and Sex ($F_{1, 75}$ = 4.157, p = 0.045, $\eta_p^2$ = 0.053). Post hoc Scheffé test revealed that females were more sensitive to the difference between the outcomes, indicating females' sensitivity to events violating their expectations regarding absence of entities (Unexpected Appearance, $M$ = 0.177, SD = 0.509; Expected Appearance, $M$ = −0.156, SD = 0.556, p = 0.003, $\eta^2$ = 0.091). In contrast, males did not appear to discriminate between expected ($M$ = 0.029, SD = 0.402) and unexpected outcomes ($M$ = 0.065, SD = 0.506, p = 0.738, $\eta^2$ = 0.002). Importantly, there was no interaction between Experiment and Outcome factors ($F_{1, 75}$ = 0.156, p = 0.856), suggesting that the effects were similar in all the three experiments targeting absence representation. When comparing the Lateralization Index to chance level (0), female ($t_{38}$ = 2.173, p = 0.036, $d$ = 0.348) but not male chicks ($t_{41}$ = 0.837, p = 0.407, $d$ = 0.128) showed a statistically significant left-eye bias when they were confronted with the appearance of an object at a location that should have been empty. These results together point to a sex-dependent abstract representation of absence in chicks, which is not attributable to a perceptual mismatch (Experiment 3) nor to the expectation of the presence of the imprinted object at a different location (Experiment 4). As Study 2 was specifically designed to investigate questions related to the nature of absence representations as reflected by the left-eye bias observed in Experiment 2, in Experiments 3 and 4 we did not aim to target overall looking times and neither we had any predictions regarding these. However, for completeness, we report these in Supplementary Material.

## Discussion

Several studies have shown that human infants (*Wynn and Chiang, 2016*; *Kaufman et al., 2003*), as well as other primates (*Amici et al., 2010*) and even domestic chicks as young as 4–5 days (*Vallortigara et al., 1998*; *Regolin et al., 2005*; *Chiandetti and Vallortigara, 2011*), represent the presence and the location of an object or agent even if it is no longer directly perceivable for them. Such a feat was proposed to be achieved by creating and maintaining an object index (*Scholl and Pylyshyn, 1999*) or an object file (*Kahneman et al., 1992*), which is dynamically linked to the location of the object and survives its occlusion. While such a tracking system can encode the presence of physical objects or animate agents at specific locations, it seems to break down at representing their absence (*Scholl and Pylyshyn, 1999*; *Cheries et al., 2008*). Indeed, previous research indicates an asymmetry between these two cognitive abilities (*Hearst, 1991*). While human adults (*Newman et al., 1980*) and pigeons (*Hearst, 1984*) readily learn the relation between the presence of a stimulus and

reinforcement, they have difficulties with recognizing the relation between the absence of a stimulus and reinforcement. Furthermore, human infants show remarkable capacities of object representation (*Carey, 2009*), but spontaneously representing the absence of an object seems to exceed their abilities (*Wynn and Chiang, 2016*; *Kaufman et al., 2003*). Thus, grasping absent entities likely exploits additional cognitive mechanisms that are beyond those recruited for representing presence.

We developed a new paradigm to investigate domestic chicks' spontaneous expectations regarding presence and absence of objects. We used two independent measurements to tackle on chicks' ability to recognize possible violations of their expectations. First, we predicted longer looking times in response to violations of expectations about presence, and potentially to violations of expectations about absence. Second, we predicted left-eye bias in response to unexpected events. In Study 1, we found that, similar to human infants (*Wynn and Chiang, 2016*; *Kaufman et al., 2003*), chicks' looking times toward expected and unexpected outcomes were differently modulated in experiments testing the representation of presence and absence of an object. However, looking time similar to the observations of human experiments with infants (*Wynn and Chiang, 2016*; *Kaufman et al., 2003*) was not sensitive to possible violations of absence representations. We also measured chicks' eye bias and predicted preferential left-eye usage in response to unexpectedly disappearing and appearing objects. This measurement was selectively sensitive only to scenes involving an object unexpectedly appearing (violating expectations about its absence) in three experiments. Thus, remarkably, we found sex-dependent evidence for young chicks' ability to represent the absence of an object in their eye usage in the three experiments targeting absence representation (Experiments 2–4). In Study 2, we aimed to ask further questions regarding the processes that may be involved in of chicks' expectation about absence. The results of Experiment 3 ruled out the possibility that the left-eye bias associated with the novelty of inspected stimuli was the result of a mismatch between information stored in perceptual memory and the outcome of the scene. Experiment 4 showed that preferential left-eye use was independent of tracking the approximate location of a specific object. Instead, chicks' representations of absence seem to involve complex mental computations and possibly complex representations, that go beyond the object tracking system. Altogether, our findings support the conclusion that female chicks encode not only the presence but also the absence of objects in their environment.

While we expected dominant left-eye usage in response to unexpected events in general, this measurement was sensitive specifically to violations of expectations of absence. We propose that the left-eye bias reflects chicks' attempt to *identify* a novel object that appeared at a location where 'nothing' was expected to be. Two arguments support this interpretation. First, while our initial prediction was left-eye bias in response to unexpected events in general, this behavior was found only for outcomes *involving an object* (Experiments 2–4), but not when the chicks were confronted with an empty space, where an object should have been present (Experiment 1). Previous studies reporting left-eye bias in domestic chicks (*Rogers and Anson, 1979*; *Dharmaretnam and Andrew, 1994*) also involved objects as stimuli. According to our knowledge, no former work found left-eye preference in domestic chicks without a physical object. This suggests that this specific behavior is closely connected to the perceptual processing of objects, which in our case were most likely social (imprinting) objects. Second, the left-eye bias was *prominent in females* compared to males, and such sex-dependent differences are congruent with findings in social discrimination experiments with chicks (*Vallortigara et al., 1990Vallortigara and Versace, 2017*; *Vallortigara and Andrew, 1994*; *Versace et al., 2017*; *Pallante et al., 2020*). It is important to note that we do not have theoretical reasons to assume that males cannot represent absence, instead, we believe that in this specific experiment, males' abilities could have been masked by a decreased attention to familiar social stimuli in the encoding phase, or by other factors. Indeed, female chicks are more interested in familiar social partners, while males are more interested in unfamiliar ones (*Vallortigara, 1992*), and these preferences are also observed in experiments using artificial social partners (*Vallortigara and Andrew, 1994*) and different sensory modalities (*Versace et al., 2017*). These preferences most likely are related to eco-ethological characteristics of this species. Specifically, in natural populations adult fowls exhibit territorial behavior, wherein single dominant cocks maintain and patrol a large territory within which a number of females live, and this sort of social organization favors the prevalence of gregarious and affiliative behaviors in females (*McBride and Foenander, 1962*; *Andrew, 1966*). Furthermore, females are more inclined to use the right eye for a conspecific and the left eye for a novel stimulus (*Dharmaretnam and Andrew, 1994*), and 8-day-old chicks prefer to look at an approaching unfamiliar object using their left eye

(*Rogers and Anson, 1979*). Based on these previous findings, the most plausible explanation of the pattern observed in the present work is that female chicks identified the unexpectedly appearing object as a novel, unfamiliar object, even when it looked identical to an object they were familiarized (and, in fact, imprinted) to. Thus, the chicks attempted to reidentify the object despite its match to the imprinted perceptual profile, possibly because they deemed it unfamiliar as its presence at a location represented as empty contradicted their expectation — the absence of objects. A further emerging question is how exactly this unexpectedly emerging object was encoded and whether similar effects would emerge for encoding the absence of nonsocial stimuli. While the present studies were not designed to directly test whether unexpectedly emerging object was perceived as an animate social individual or an inanimate object, further studies might focus on this question by comparing different types of stimuli (social and nonsocial) and also investigate whether males and females would react differently to such items.

One might wonder whether the left-eye bias might be a signature of creating a new object representation (from a default state of no specific representation). While we believe that our results do imply that chicks have formed a new object representation in these situations, we think this process itself cannot explain our main results. First, earlier studies (e.g., *Dharmaretnam and Andrew, 1994*; *Rogers and Anson, 1979*; *Rosa Salva et al., 2007* ) documented eye biases (left or right) that were dependent on the identity (and not on the presence) of a stimulus. Thus, we think it is more likely that the left-eye bias we found in the present study also reflects object identification processes, rather than simply establishing an object representation. Moreover, only females discriminated reliably between expected and unexpected appearances of objects in our studies. However, we have no reasons to assume that basic visual information processing, like forming an object representation would elicit pronounced sex differences. Last, the timescale of our measurement period (30 s) was designed to target not simply the establishment of object representations, but the further processes performed on these representations (e.g., identification, comparing two representations, etc.). The long measurement period allowed us to obtain responses to the mismatch between expectations and outcomes that go beyond representing the presence of objects, which is a fast and automatic process usually demonstrated at the millisecond level. Thus, we believe that the explanation we offer is the most plausible one: the left-eye bias observed in the present study is likely a signature of chicks' attempt to identify the unexpectedly appearing object, given that its presence contradicts their expectation of absence of objects.

While our data provide evidence that 8-day-old chicks can encode the absence of objects — a representation that goes beyond perceiving empty space — the nature of this capacity was not directly addressed here. However, we consider important briefly discussing the theoretical possibilities to embed our work in the literature and to stimulate further research. Following Nieder's taxonomy (*Nieder, 2016*), the chicks in our study might have encoded absence either in a numerical or a categorical representational format. Regarding the numerical format, the approximate number system provides numerical representations to pre- and nonlinguistic creatures as well (*Dehaene, 1997*; *Haun et al., 2011*; *Brannon and Merritt, 2011*; *Brannon and Roitman, 2003*). Absence could be represented by this system as an approximate numerical value of less than 1. The other possible way to capture absence is via categorical representations, which are likely recruited in perceptual decision tasks to contrast the absence of an object to its presence (*Merten and Nieder, 2012*). However, the exact coding mechanism underlying categorical representation of absence and presence was not previously addressed. Whether specific pairs of contrary concepts (i.e., presence/absence) or general category-forming processes support such abilities is the potential target of future research.

In our experiments, chicks might have encoded either the approximate number of objects behind the screen (~1 or ~0), or formed a categorical representation of the presence or absence of objects behind the screen. Nevertheless, one aspect of our results supports more the option of a categorical encoding of absence: in Experiments 2–4, the female chicks identified the imprinting object as an unfamiliar one, which suggests that they relied on strong evidence regarding its identity (i.e., that it was not the imprinting object). Representing the approximate number of objects behind the screen (i.e., 'roughly zero') is unlikely to lead to such a conclusion. In contrast, a categorical expectation about having no object at a particular location would support identifying the object as a novel 1. Furthermore, our results indicate different cognitive processes underlying the representation of presence and absence. Such an asymmetric behavioral pattern has never been found in studies in which responding

to 0 or 1 object relied on the approximate number system in numerical tasks. In contrast, studies likely evoking a categorical encoding of presence and absence consistently reported such asymmetries (*Newman et al., 1980*; *Hearst, 1984*; *Hearst, 1991*).

Importantly, in former studies, animals that managed to use absence information performed experimental tasks involving hundreds of trials. In contrast, the chicks in our study did not receive any training or practice with forming categories of presence or absence, nevertheless their performance indicated their readiness to process this type of information. The fact that 8-day-old chicks with scarce visual experience were able to do so points to the fundamental nature of such representations. Such an ability could serve not only foraging and safety purposes (i.e., encoding and maintaining the absence of food or predators at a specific location) but could also contribute to the reidentification of entities in the animal's environment, and in humans also supports the development of abstract concepts, such 'nothing'.

## Materials and methods

### Subjects and rearing conditions

Newborn chicks of the Ross 308 strain (broiler with fast growth rate) were collected from the incubator a few hours after hatching and they were housed individually in standard conditions in rectangular-shaped home cages (28 cm wide × 40 cm high × 32 cm deep) with a small circular opening on the front side (8 cm high, 2.7 cm diameter). Chicks could insert their head in this hole and look outside, and in this way they became familiarized with protruding the head from a window. Each chick shared the home cage with a red cylinder-shaped object (3 cm wide × 5.5 cm high) suspended centrally by a fine thread at about his or her eye level. The red object served as an imprinting object. Water and food were available ad libitum.

### Apparatus

Training and testing took place in a separate room close to the rearing room. The experimental apparatus consisted of a white circular arena (diameter 66 cm × 50 cm high), a screen (15.5 cm wide × 13 cm high), and a confining cylinder (10 cm wide × 25 cm high) in which the chick was placed during the training and testing sessions. The confining cylinder had a small circular opening facing the center of the arena. This opening had the same dimension and position as the windows on the home cages (8 cm high, 2.7 cm diameter). The screen was made of a plastic opaque blue sheet and it was placed at 30 cm from the closest part of the confining cylinder. The experimenter could rotate the screen upward (to reach a vertical position) and downward (leaning it forward on the apparatus floor) from above the experimental arena to hide or reveal the space behind the screen. Behind the screen, a camouflaged sliding door in the floor made possible for the experimenter to secretly remove or place the target object behind the screen out of the chicks' view. A red object identical to the imprinting object was used during the training and the test. The experimenter could move the object within the arena with a fine thread held from above. A video camera was placed above the confining cylinder and recorded the whole test session.

### Procedure

#### Familiarization:

On day 4 or 5 after hatching all chicks underwent a short familiarization session to get acquainted with the apparatus. Chicks were gently placed inside the confining cylinder and they could put out the head through the window. The red object was moved between the confining cylinder and the screen (which was in vertical position), but importantly the object never went behind the screen, thus chicks did not experience the occlusion of the object in this phase. Mealworms were placed on the top of the red object so if the chick put the head out of the cylinder and the red object was at reachable distance the chick could get the mealworm. After the chick ate the mealworm on the top of the red object, the object was removed from the arena and the mealworm was replaced. The red object was always inserted and removed centrally at the back of the experimental arena, opposite to the confining cylinder. The familiarization phase lasted 10 min for each chick. Chicks not putting out their head from the confining cylinder, and thus not seeing any of the events or not taking the mealworm were excluded from the experiment (n=21).

## Test

On day 8 after hatching chicks participated in the test of one of the four experiments. We tested 8-day-old chicks, because piloting of the paradigm showed that chicks at this age (but not younger) are more likely to put out their heads through the window of the confining cylinder, and to provide sufficient data for measurement. Every test session started with a short warm-up period, in which chicks were presented with the upward/downward rotating movement of the screen and the left/right movements of the red object. First, the screen was moved upwards and downwards three times. After the last movement, the screen remained in vertical position. Then the object was moved from near the window of the confining cylinder toward the screen and then behind the screen, and thus disappearing from the chick's sight. Afterwards, the object was moved back from behind the screen to the window of the confining cylinder. This action was repeated once on the left and once on the right side. Afterwards, the screen was rotated downwards to horizontal position, and the object was moved on similar trajectories. After the warm-up period the test events followed. In all experiments trials were counterbalanced in an ABBA order, in a total of four test trials (where As and Bs stand for the two outcomes, counterbalanced across participants). The outcome was presented for approximately 30 s, with a small variance due to the natural variation of the movement of the experimenter while rotating upwards the screen at the end of the trial. To rule out the potential effect of this variance, we coded the length of trials and we investigated its effect on the dependent variables. First of all, we did not find significant difference between the length of trials in the two experimental conditions (Expected condition, $M$ = 31.34, SD = 1.81; Unexpected condition, $M$ = 31.15, SD = 1.89, Mann–Whitney $U$ test, $U$ = 22.56, p = 0.35). Moreover, we did not find significant correlation either between the length of trials and Looking Time ($r$(434) = 0.013, p = 0.793) or between the length of trials and Lateralization Index ($r$(434) = −0.003, p = 0.949). Chicks not putting out their heads from the confining cylinder during the presentation of the test events or through the full duration of presenting the outcome could not provide any data and were excluded from the studies (4 chicks in Experiment 1, 5 in Experiment 2, 4 in Experiment 3 and 2 in Experiment 4).

## Experiment 1 (Encoding Presence)

In the Unexpected Disappearance condition, the object was first placed behind the screen but then it was secretly removed from the arena through a sliding door hidden behind the screen, in a way that chicks could not see the removal. In the outcome phase, the screen was dropped and it revealed the empty space behind the screen. In the Expected Disappearance condition, the object was removed from the arena in the full view of the chick. In order to equalize the amount of noise and to approximately even up the time passing before the outcome, the sliding door was moved back and forth similar to the Unexpected Disappearance condition. In the outcome phase, the screen was dropped and it revealed the empty space behind the occlude (see *Video 1*). The movement of the object was counterbalanced within subjects: it was removed or moved behind the screen once from left and once from right in both conditions across the four trials, order counterbalanced.

## Experiment 2 (Encoding Absence)

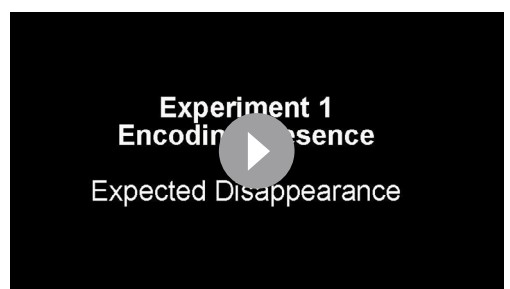

**Video 1.** Test events in Experiment 1 (Encoding Presence). The chick is presented with the Expected and Unexpected Disappearance of the imprinting object (example video from the test set).
https://elifesciences.org/articles/67208/figures#video1

In the Unexpected Appearance condition, the object was visibly removed from the arena and then the screen was positioned in vertical position (hiding the space behind it). Afterwards, the object was secretly placed behind the screen, through the sliding door, in a way that chicks could not see the placement. In the outcome phase the screen was dropped and the object was revealed. In the Expected Appearance condition, the object was moved toward and placed behind the lowered screen. Then, the screen was raised in vertical position – covering the object – and the sliding door was moved back and forth similar to the Unexpected Appearance condition. At the

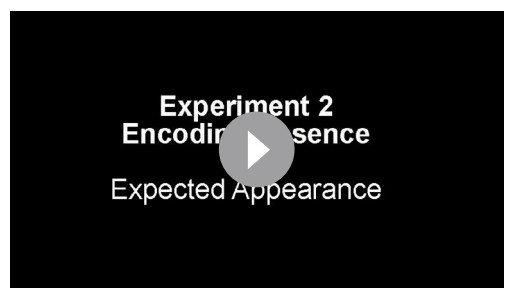

**Video 2.** Test events in Experiment 2 (Encoding Absence). The chick is presented with the expected and unexpected appearance of the imprinting object (example video from the test set).
https://elifesciences.org/articles/67208/figures#video2

end of the trial, the screen was dropped revealing the presence of the object (see *Video 2*).

## Experiment 3 (Absence vs. Perceptual Comparison)

Procedure was similar to Experiment 2 except the following changes. In the warm-up phase, the screen was moved upwards and downwards three times, stopping in a horizontal position while the object was moved left and right twice. Then the same movement was repeated with the screen in vertical position. In Unexpected Appearance condition the object was moved behind the screen and then it was removed from the arena visibly to the chick. Before the end of trial, the object was secretly placed behind the screen, through the secret door. At the end of the trial, the screen was dropped and it revealed the object. In Expected Appearance condition, first the object was moved behind the screen and then to equalize the movements in the two conditions, it was moved halfway within the space between the screen and the edge of the arena to reveal it again for the chick, and afterwards it was moved back and placed again behind the screen. At the end of the trial the screen was dropped revealing the object (see *Video 3*).

## Experiment 4 (Absence vs. Tracking)

In this experiment, we used two objects both in the familiarization and in the test session. One object – just as in the other three experiments – was identical to the red imprinting object provided for all chicks in their home cages from the first day of life. The other object (hereafter the green object) was first introduced during the familiarization. This object had the same shape and size as the red object but it was green with yellow stripes on the bottom and top part. During the 10 min of the training session two objects were alternated. In the test session, the pretest phase was the same as in Experiment 3 with the exception that the two objects were presented in turns; once the red was moved left and right and then the green object was moved to left and right while the screen was in horizontal position and the same movements were presented while the screen was in vertical position.

The expected events were similar to the events in Experiment 3. In the Unexpected Appearance condition, the green object was moved behind the screen and then it was removed from the arena in the view of the chick. Afterwards, the red object was secretly placed behind the screen through the sliding door. At the end of the trial the screen was dropped uncovering the red object (see *Video 4*).

## Data analyses

We derived two dependent variables from the coded values: Looking Time and a Lateralization Index. Looking Time was the sum of the left, the right, and the binocular eye usage during the 30 s. Lateralization Index was calculated as the ratio of the difference between left- and right-eye usage compared to the total of left- and right-eye usage (i.e., (Left − Right)/(Left + Right)).

The looking behavior of the chicks was coded offline, analyzing the position of the head when it was outside of the confining cylinder (minimum the whole beak of the bird had to be outside) toward to the region of interest (the test outcome: object being present or absent). Every trial was coded frame-by-frame using a transparent plastic sheet displaying the coding angles to determine the visual hemifield used to look at the outcomes.

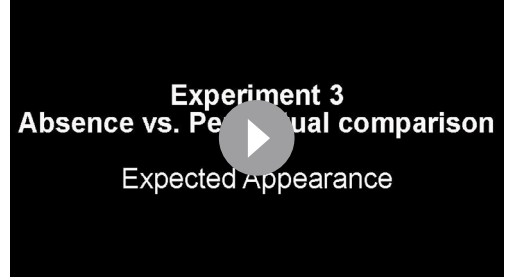

**Video 3.** Test events in Experiment 3 (Absence vs. Perceptual Comparison). The chick is presented with the expected and unexpected appearance of the imprinting object (example video from the test set).
https://elifesciences.org/articles/67208/figures#video3

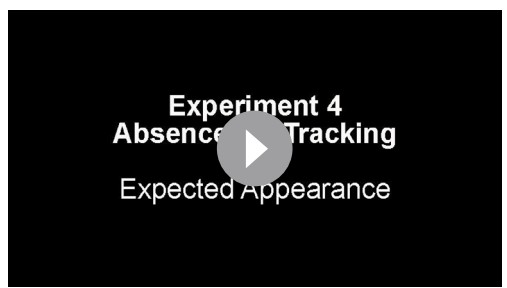

**Video 4.** Test events in Experiment 4 (Absence vs. Tracking). The chick is presented with the expected and unexpected appearance of the imprinting object (example video from the test set).
https://elifesciences.org/articles/67208/figures#video4

The midline was positioned on the centerline of the beak and right monocular field was defined as 15–135° from the middle, left monocular field was defined as −15/−135° from the middle and binocular field was specified as ±15° from the middle. Fields of view falling outside of these values, looking down (when the beak was not visible) and looking up (one eye was positioned up) were discarded from the data analyses. In Experiment 1, where no object was present in the outcome phase of either condition, the region of interest was defined as the space that was earlier covered by the screen (lasting till edges of the lowered screen). In Experiments 2–4, where an object was present in the outcome phase, the region of interest was defined as the boundaries of the object beyond the lowered screen. The reliability of coding was strengthened via a comparison between the originally coded data and a second data set of 10% of the participants coded by a blind coder. The two data sets highly correlated ($r = 0.98$).

Looking time data of Experiments 3 and 4 were not analysed, as these experiments were specifically designed to investigate the left-eye bias observed in Experiment 2. However, for completeness, we now report the means and the standard error values of the looking time data (merged and separately for males and females) for all experiments and the analyses of the looking time data of Experiments 3 and 4 in Supplementary Information (see *Supplementary file 1*).

## Acknowledgements

This project was supported by funding from the European Research Council (ERC) Advanced Grants under the European Union's Horizon 2020 research and innovation program (grant agreement no. 833,504 SPANUMBRA) and PRIN 2017 ERC-SH4-A (2017PSRHPZ) to GV, European Union's Seventh Framework Programme (FP7/2007-2013) ERC Grant (284236), REPCOLLAB to ÁMK, and European Union's Horizon 2020 Research and Innovation Programme ERC Grant (639840), PreLog to ET.

## Additional information

### Funding

| Funder | Grant reference number | Author |
|---|---|---|
| European Research Council | Advanced Grant 833504 - SPANUMBRA | Giorgio Vallortigara |
| PRIN 2017 ERC-SH4-A | 2017PSRHPZ | Giorgio Vallortigara |
| European Research Council | Starting Grant 639840 - PRELOG | Ernő Téglás |
| European Research Council | Starting Grant 284236 - REPCOLLAB | Ágnes Melinda Kovács |

The funders had no role in study design, data collection, and interpretation, or the decision to submit the work for publication.

### Author contributions

Eszter Szabó, Conceptualization, Data curation, Formal analysis, Visualization, Writing - original draft; Cinzia Chiandetti, Conceptualization, Data curation, Methodology; Ernő Téglás, Gergely Csibra, Conceptualization, Writing - review and editing; Elisabetta Versace, Conceptualization, Methodology, Writing - review and editing; Ágnes Melinda Kovács, Giorgio Vallortigara, Conceptualization, Funding acquisition, Supervision, Writing - review and editing

## Author ORCIDs

Eszter Szabó ⬤ http://orcid.org/0000-0002-7021-7112
Cinzia Chiandetti ⬤ http://orcid.org/0000-0002-7774-6068
Elisabetta Versace ⬤ http://orcid.org/0000-0003-4578-1851
Gergely Csibra ⬤ http://orcid.org/0000-0002-7044-3056
Ágnes Melinda Kovács ⬤ http://orcid.org/0000-0003-3452-7727
Giorgio Vallortigara ⬤ http://orcid.org/0000-0001-8192-9062

## Ethics

All experiments comply with the current Italian and European Community laws for the ethical treatment of animals and the experimental procedures were approved by the Ethical Committee of University of Trento and licensed by the Italian Health Ministry (permit number 1138/2015 PR).

## Decision letter and Author response

Decision letter https://doi.org/10.7554/eLife.67208.sa1
Author response https://doi.org/10.7554/eLife.67208.sa2

---

## Additional files

### Supplementary files

• Transparent reporting form
• Supplementary file 1. Looking time data in Experiment 1-4.

### Data availability

Data are available in the Harvard Dataverse at https://doi.org/10.7910/DVN/RBDYKW.

The following dataset was generated:

| Author(s) | Year | Dataset title | Dataset URL | Database and Identifier |
|---|---|---|---|---|
| Szabó E | 2021 | Data for "Young domestic chicks spontaneously represent the absence of objects" manuscript | https://doi.org/10.7910/DVN/RBDYKW | Harvard Dataverse, 10.7910/DVN/RBDYKW |

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
