## [Editor Report]

The research detailed in this manuscript investigates whether young chicks represent the absence of objects. This work is important to multiple fields of inquiry, such as ethology and neuroscience, and is the first time that this ability has been demonstrated to be exhibited spontaneously, as opposed to after many trials of experience.

---

## [Decision Letter]

**Decision letter after peer review:**

Thank you for submitting your article "Young domestic chicks spontaneously represent the absence of objects" for consideration by *eLife*. Your article has been reviewed by 3 peer reviewers, including Kerry Jordan as Reviewing Editor and Reviewer #2, and the evaluation has been overseen by Christian Rutz as the Senior Editor. All of the Reviewers are familiar with and have published in numerical cognition, animal behavior, and comparative development. The following individual involved in review of your submission has agreed to reveal their identity: Pier Francesco Ferrari (Reviewer #1)

All three Reviewers agree that this work has potential, and the investigated topic is innovative. We do not believe that additional experiments are required. As you will note from the Reviewers' comments, concerns are primarily limited to presentation and the substantial amount of clarification still needed.

Essential revisions:

(1) Clarity needed regarding interpretations of results. I would recommend, as do the Reviewers, especially working to clarify the results and interpretations regarding sex differences in the ability of chicks to represent the absence of objects via side eye bias. As noted by Reviewer 1, you should carefully and completely address why your results/interpretation of this diverges from previous literature. Multiple reviewers also note that this general concept and analysis plan should be introduced earlier in the manuscript.

(2) Why this age chick was chosen as the test age. All 3 reviewers questioned this.

I believe these issues can be sufficiently addressed in a revision. A revision is thus being requested of this manuscript. In addition to the main points described above, I also suggest answering all reviewer questions that can be answered with the current data/experiment.

We each recognize the great deal of work and effort that went into the original manuscript, and of course will still be put in if you decide to undertake the revision. If you have any questions about the reviews or my decision, please feel free to contact me.

Thank you for considering *eLife* as an outlet for your work.

*Reviewer #1 (Recommendations for the authors):*

The methodologies of the experiments are well carried out and all the information contained in the paper allows other researchers to possibly replicate the study. Minor, but nevertheless important details, could be missing if a naïve observer has no possibility to visualize the experiments. Thus, video material can be useful in order to verify, for example, how can the experimenter introduce or remove objects from the chick's visual field, without providing additional information, which, by mistake, could have been ignored by the experimenters. This must be part of supporting material.

The main finding of the study is that chicks encode spontaneously the absence of objects. This was the original hypothesis tested here. Part of the results support such conclusion.

General looking time show that there is more looking for the unexpected disappearance (Exp. 1A) but not the unexpected appearance (Exp. 1B). Without looking at left-right eye bias, this result show that the typical behavioral marker of violation expectancy is present for Exp. 1A but not 1B. The following analyses however show the left-eye bias for encoding absence. What is not clear is whether in the absence of the main marker of violation expectancy (i.e. looking time) other behavioral markers could be considered equally valid as strong marker of violation expectancy. To me the left-eye bias is an additional information that should complement the data of violation expectancy based on looking time. This conceptual issue should be adequately addressed and better justified.

– Experiments 3 and 4 should in any case report the statistical analysis and results of looking time in order to see how robust is the effect in order to support the conclusions.

– By looking at a few videos of the experiments, I wonder if, at the beginning of phase 3 (i.e. when the screen is lowered) the initial position of the chick's head could drive a persistent head posture which favors a specific eye bias. Chicks often follow with one eye the appearance and disappearance of objects. Did authors balance the direction from where objects were moved into the arena? This information should be mentioned in the method section. If this has not been controlled, then a specific analysis should assess for this possible confounding effect on the eye bias.

– In terms of results, the effects found are present only if the analysis is separated for left and right eye, showing a clear left-eye bias. Moreover, the result is only present in females, thus raising some concerns about the robustness of this phenomenon and its adaptive value.

If females' left-eye bias is a chicks' signature to identify the unexpectedly appearing object, given its survival importance, such early competence should be present in males' chicks too.

Much of the discussion tries to explain such differences but the arguments are sometimes incoherent (see below).

– Most of the discussion is devoted in fact to the explanation of why such biases are present only in females and the arguments are not totally convincing (i.e. females are more interested to social partners). The stimuli used here are inanimate objects. Previous data on left/right bias in relation to social and nonsocial stimuli are in line with current findings (see Dharmaretnam and Andrew, 1994) but then the argument of sociality cannot explain the sex differences since females have left-eye bias to an inanimate object. Current findings sometimes are in agreement with some but not all the previous literature.

– In relation to the previous point, in the series of experiments presented here, authors did not test different types of stimuli (social vs nonsocial). This experiment could be relevant to support authors' argument of females being more interested to social stimuli.

Other comments

In general, I found the introduction well written but since the behavioral analysis has been mainly set around the left-right eye use, the rationale of the experiment has not been sufficiently elaborated in the introduction. No predictions have been put forward to justify the statistical approach adopted by the authors. This leaves the reader a bit confused while reading the result section.

I recommend authors to shorten the general introduction and to develop it more around the laterality issue and its relevance for the current study. Predictions should be better spelled out.

– The left-eye preference without physical object is present in females but not males, and it has been interpreted within the framework of female's more efficient capacity of social discrimination. There are few logical passages which require to be better explained.

The first is that the imprinted object used in the current experiments is not a social stimulus. Therefore, one should test whether this left-eye bias is stronger for social vs nonsocial objects. Secondly, if computing absence is not domain specific (i.e. social), but a general capacity of the species, why males should not have it?

Furthermore, the work by Darmaretnam and Andrew shows that females are more inclined to use the right eye for a conspecific which contradicts previous interpretation (on sex differences). In brief, the first attempt to interpret data is aimed at justifying why females have the left-eye bias (p.13), but then the same argument does not hold because the right-eye bias is more prominent for conspecifics and not for objects. I think authors should be coherent and avoid twisting the arguments because this part of the discussion is rather confused.

– One striking difference with humans, is that representing absence of objects emerge between one to two years. In chicks, this is present at 8 days of age. I am not sure it makes sense to compare developmentally chicks and humans. Such comparison could be extremely useful to understand whether different behavioral responses to violations are based on the same (homologous or analogous) mechanisms, or whether in chicks perceptual and cognitive processes are specializations evolved as specific adaptations to their environment. Overall, honey-bees has the concept of "empty sets" despite their nervous system is substantially different from that of other vertebrates. I think that it would be useful to have some conceptualizations of how the behavioral phenomenon described here in chicks is based on mechanisms that could have evolved independently in other species in order to accomplish simple cognitive processes useful in specific domains (e.g. foraging, tracking conspecifics). The linguistic concept of "zero" could be an extension of such basic components.

– At the end of the abstract it is suggested that chicks form "thoughts" about the absence. I think this term could raise some doubts among scholars (not only linguists), and I think its use here is neither necessary nor appropriate.

– Figure 2B and Figure 4 probably contain asterisks to mark differences. They are almost invisible to the reader. If that is the case these figures should be remade and the legend should explain what the asterisks indicate.

– Along the text (Figures 1 and 3) authors used the terms 'encoding' and 'computing'. Are these terms, according to authors, indicating different concepts? If yes, these should be explained in the text, otherwise please be consistent with one of them.

– I found the last third of the discussion (from: "While our data provide……" until the end) rather speculative. It should be shortened into one paragraph focusing on the possibility of a categorical encoding of absence in the species.

*Reviewer #2 (Recommendations for the authors):*

Before publication, I suggest the authors clarify answers to the following specific queries:

Introduction

On the first page, the authors state that "by the age of 5 children seem to successfully operate with counterintuitive concepts like zero and nothing". The authors need to specify why this is counterintuitive, or remove such a statement (especially when on page 2 they say, "Things can be present or absent, yet how these intuitively simple opposing categories are formed and encoded is largely unexplored"). Maybe introducing the idea of contrary concepts here, instead of leaving that for the Discussion, could be helpful?

Previous research with rooks is mentioned briefly in one sentence near the end of the Introduction. But if this has already been done in another bird species, how does doing this in chicks further significantly advance knowledge? Or, was the rook research with adult birds rather than neonates?

A little more background review on eye usage with respect to novelty would be useful. What is the theory behind why this occurs? Is it specific to birds, and if so, why?

Methods

Why were 8-day-old chicks the age chosen? Is there precedent in previous research for the eye bias to be present starting then (as hinted at in the Discussion), is this the crucial time period after Imprinting/Familiarization for Test for some particular reason, etc?

Results

It is interesting sex was a factor in the ANOVAs, and did sometimes have an effect. Was this predicted? It is great that later in the Discussion the authors speculate why there may be sex-dependent representation of absence in chicks, but I think there needs to be background information included earlier so the reader can understand why this was part of the Analysis plan. I am also just generally confused about this speculation. Is the imprinting object not a social/conspecific equivalent object? How then would we really predict (and interpret!), based on previous literature, whether to expect a left or right side bias? Is it perhaps a weighted prediction (e.g., the weight of importance for predicting absence of objects in females is more important than the weight of right-eye-bias predictions…but for males, these weights may be more evenly distributed between sides?) This needs to be clarified.

In Study 2, why are there no untransformed mean looking time results given in the text, as there were for Study 1?

Discussion

On page 12, the authors should clarify whether "Thus, remarkably, we found sex-dependent evidence for young chicks' ability to represent the absence of an object in their eye usage" applies to all Studies, or just a subset.

Overall, this is subject matter interesting to readers of *eLife*. I think this could be suitable for publication if these concerns were addressed well.

*Reviewer #3 (Recommendations for the authors):*

The experiments are well conducted and discussed, thus I have very limited suggestions.

– I suggest the Authors clarify which are the cognitive mechanisms underlying the representation of the absence of objects. In the initial part of the introduction, the approximate number system and the object tracking system are presented as the cognitive systems underlying numerical comprehension in animals. Nevertheless, it is unclear which system was considered responsible for the representation of absence. Moreover, I suggest explaining why "chicks' representations of absence seem to involve complex mental computations and possibly complex representations, that go beyond the object tracking system". This system is devoted to object representation, thus it seems the most suitable to be triggered by the presentation of an imprinting object. This system can estimate the number of perceived objects (also in this animal model), possibly starting from an empty set, thus it is unclear why it cannot be responsible for a representation of the absence of objects. Considering the introduction opens with an illustration of the cognitive systems underlying numerical cognition, I suggest integrating these findings with the previous literature showing numerical competences and the respective cognitive mechanisms in the animal model used in this study. Moreover, since this study is based on the prerequisite of object permanence, it would be interesting to discuss this evidence in more detail.

– I also suggest to better distinguish the concept of zero and the representation of absence of objects in the introduction.

– Scientific evidence on numerical competences in domestic chicks is mainly provided in the very first 5 days of life. Thus, I suggest specifying why here, chicks were tested at 8 days of life. Is this choice related to the employed paradigm or to an assumed underling cognitive development necessary for the chicks to grasp this concept?

– It is unclear to me why Experiment 1 is labeled as "encoding presence" and Experiment 2 "encoding absence".

– The results of the comparison between the Expected disappearance (Exp 1) and unexpected appearance (Exp 2) might merit a deeper discussion. They share the initial part of the presentation of the stimulus, but differ for the outcomes (the presence or the absence of the object once the panel was lowered).

– I suggest reporting all the effect sizes and to modulate the discussion on the basis of their values.

– Specify how the sample size was calculated.

– Italicize all species names in the references.

---

## [Author Response]

Essential revisions:(1) Clarity needed regarding interpretations of results. I would recommend, as do the Reviewers, especially working to clarify the results and interpretations regarding sex differences in the ability of chicks to represent the absence of objects via side eye bias. As noted by Reviewer 1, you should carefully and completely address why your results/interpretation of this diverges from previous literature. Multiple reviewers also note that this general concept and analysis plan should be introduced earlier in the manuscript.

Following this suggestion we have now performed a series of changes aiming to clarify the results and interpretations regarding the sex differences obtained, which are described in detail at the responses to reviewers. In particular: 1. we have streamlined the introduction, where we have integrated earlier findings with chicks relying on eye lateralization as well as discuss the studies which have found sex differences and we have described in more detail the conceptual motivation of the study and the analysis plan, and we also address in the discussion how our results and interpretation relate to previous literature.

(2) Why this age chick was chosen as the test age. All 3 reviewers questioned this.

We have now specified the two main reasons: the procedure requires a solid imprinting that motivates the chicks to search for the imprinting object putting their heads in the window in a confined cylinder. Pilot data showed that chicks don’t have this motivation before 4-5 days of life. Moreover, Dharmaretnamn and Andrew (1994) showed that at day 8, chicks are strongly lateralised in eye use with the imprinting and novel stimuli. We added the following details to the manuscript at page 4: “The age was determined by the specific paradigm we used in the present work (see the details of imprinting and familiarization with the apparatus in the Materials and methods) and the dependent measures we targeted (looking time and lateralization index). This is also the age when chicks were found to show strong lateralized responses to familiar and unfamiliar objects (Dharmaretnam and Andrew 1994).”

Reviewer #1 (Recommendations for the authors):The methodologies of the experiments are well carried out and all the information contained in the paper allows other researchers to possibly replicate the study. Minor, but nevertheless important details, could be missing if a naïve observer has no possibility to visualize the experiments. Thus, video material can be useful in order to verify, for example, how can the experimenter introduce or remove objects from the chick's visual field, without providing additional information, which, by mistake, could have been ignored by the experimenters. This must be part of supporting material.

We would like to thank Reviewer 1 for pointing this out, we are happy to add to the supplementary material the suggested video material.

The main finding of the study is that chicks encode spontaneously the absence of objects. This was the original hypothesis tested here. Part of the results support such conclusion.General looking time show that there is more looking for the unexpected disappearance (Exp. 1A) but not the unexpected appearance (Exp. 1B). Without looking at left-right eye bias, this result show that the typical behavioral marker of violation expectancy is present for Exp. 1A but not 1B. The following analyses however show the left-eye bias for encoding absence. What is not clear is whether in the absence of the main marker of violation expectancy (i.e. looking time) other behavioral markers could be considered equally valid as strong marker of violation expectancy. To me the left-eye bias is an additional information that should complement the data of violation expectancy based on looking time. This conceptual issue should be adequately addressed and better justified.

Here we would like to clarify that we used two statistically independent measures: the overall looking time and the lateralization index. In fact, if a chick looks 10 seconds to Outcome 1, this is unrelated to the value on the lateralization index (that can be positive or negative or 0). Hence, the lateralization index is independent from looking time (they are not complementary measures).

Second, the left-eye-bias in our case does not seem to be a general marker of violation of expectancy, as it seemed to be specific for the experiments targeting absence (Exp 2-3-4) but not for the one targeting presence (Exp 1). Below we describe in detail our reasoning for using these two measures, and we have also clarified this in the manuscript.

Hereby we would like to unpack the reasoning behind using these two different and independent measures and to provide more details about the overall looking times and the lateralization index.

We measured looking times, similarly to how it is done in infancy research. However, we were aware from the start that such looking time paradigms with infants did not find evidence for encoding absence (Wynn and Chiang, 1998; Kaufmann et al., 1993, 1995). Moreover, looking time is not a frequently used measurement in avian species. This measure was used in an avian species only once to investigate a very different question, specifically whether rooks understand simple physical relations of support (see Bird and Emery, 2010).

Therefore, we used a second measure, the lateralization index. This measure is commonly used with chicks and other avian and non-avian species with laterally-placed eyes to investigate different processes. In particular, we relied on studies with chicks that have found left-eye bias for novel outcomes (Rogers and Anson, 1979; Dharmaretnam and Andrew, 1994).

In the looking time measurement, we did not find evidence supporting chicks’ understanding of absence, although we found an interaction between looking patterns in the experiments targeting Presence and Absence representations. There may be several explanations why the overall looking times did not reflected sensitivity to violation of expectation of absence. For instance, it is possible that overall looking time measurements are more sensitive to presence, rather than absence violations in different populations (e.g., for infants: Wynn and Chiang, 1998; Kufmann et al., 1993, 1995). In any case, it should be noted that this was the very first reported looking time study with chicks that we are aware of, and we should be careful in interpreting the non-discriminative overall looking time.

However, in the lateralization index we consistently observed sex dependent left-eye bias in response to unexpected appearance of the object that could be encoded as being absent in three experiments (Exp. 2-3-4 designed to test chicks’ ability to represent the absence of objects), but not when witnessing its expected appearance. Note that this eye bias did not emerge in Experiment 1, that targeted presence representations, and we compared an expected disappearance outcome with the unexpected disappearance of an object. Therefore, we have argued that this effect does not simply reflect distinguishing an expected outcome from an unexpected one (that would be a general a violation of expectation marker), but instead it relates to processes recruited by a representation of absence.

We propose that the left-eye bias emerged in response to the unexpected appearance of the object (but not to its expected appearance) is related to the identification and encoding mechanisms involved in processing a potentially new object. Likely, the test outcome object was encoded as novel object (although it looked identical to the imprinting object), and this could happen only if the imprinting object was previously encoded as being absent at that specific location (Exp 2,3) or no object was expected at that location (Exp 4).

We modified the discussion at page 11-12:

“We used two independent measurements to tackle on chicks’ ability to recognize possible violations of their expectations. First, we predicted longer looking times in response to violations of expectations about presence, and potentially to violations of expectations about absence. Second, we predicted left-eye bias in response to unexpected events. In Study 1, we found that, similarly to human infants (Wynn and Chiang, 1998; Kaufman et al., 2003), chicks’ looking times towards expected and unexpected outcomes were differently modulated in experiments testing the representation of presence and absence of an object. However, looking time similarly to the observations of human experiments with infants (Wynn and Chiang, 1998; Kaufman et al., 2003) was not sensitive to possible violations of absence representations. We also measured chicks’ eye bias and predicted preferential left-eye usage in response to unexpectedly disappearing and appearing objects. This measurement was selectively sensitive only to scenes involving an object unexpectedly appearing (violating expectations about its absence) in three experiments.

Thus, remarkably, we found sex-dependent evidence for young chicks’ ability to represent the absence of an object in their eye usage in the three experiments targeting absence representation (Experiment 2, 3, 4). In Study 2 we aimed to ask further questions regarding the processes that may be involved in of chicks’ expectation about absence. The results of Experiment 3 ruled out the possibility that the left eye bias associated with the novelty of inspected stimuli was the result of a mismatch between information stored in perceptual memory and the outcome of the scene. Experiment 4 showed that preferential left-eye use was independent of tracking the approximate location of a specific object. Instead, chicks’ representations of absence seem to involve complex mental computations and possibly complex representations, that go beyond the object tracking system. Altogether, our findings support the conclusion that female chicks encode not only the presence but also the absence of objects in their environment.

While we expected dominant left eye usage in response to unexpected events in general, this measurement was sensitive specifically to violations of expectations of absence.”

– Experiments 3 and 4 should in any case report the statistical analysis and results of looking time in order to see how robust is the effect in order to support the conclusions.

Experiments 3 and 4 investigate the possible boundary conditions of chicks’ absence representations as reflected by the left eye bias observed in Experiment 2, and asked further questions about the underlying processes. Based on Study 1, we did not expect any effects in the overall looking time measurements, thus we did not target this measure in Experiments 3 and 4. In hindsight, however, we agree that the readers may benefit from this information as well. Thus, we now report the means and the standard error values of the analyses of the looking time data (merged and separately for males and females) for all experiments in the supplementary information. However, we would like to note that our conclusions are based on the lateralization index measurement, that is independent of the overall looking time. We added the following to the manuscript at page 10-11:

“As Study 2 was specifically designed to investigate questions related to the nature of absence representations as reflected by the left eye bias observed in Experiment 2, in Experiment 3 and 4 we did not aim to target overall looking times and neither we had any predictions regarding these. However, for completeness, we report these in the Supplementary Material.”

– By looking at a few videos of the experiments, I wonder if, at the beginning of phase 3 (i.e. when the screen is lowered) the initial position of the chick's head could drive a persistent head posture which favors a specific eye bias. Chicks often follow with one eye the appearance and disappearance of objects. Did authors balance the direction from where objects were moved into the arena? This information should be mentioned in the method section. If this has not been controlled, then a specific analysis should assess for this possible confounding effect on the eye bias.

We did counterbalance the side of the crucial actions (objects leaving/appearing) within participants. We reported the counterbalancing in the Procedure on page 17 of the submitted manuscript:

“The movement of the object was counterbalanced within subjects: it was removed or moved behind the screen once from left and once from right in both conditions across the four trials, order counterbalanced.”

– In terms of results, the effects found are present only if the analysis is separated for left and right eye, showing a clear left-eye bias. Moreover, the result is only present in females, thus raising some concerns about the robustness of this phenomenon and its adaptive value.If females' left-eye bias is a chicks' signature to identify the unexpectedly appearing object, given its survival importance, such early competence should be present in males' chicks too.Much of the discussion tries to explain such differences but the arguments are sometimes incoherent (see below).– Most of the discussion is devoted in fact to the explanation of why such biases are present only in females and the arguments are not totally convincing (i.e. females are more interested to social partners). The stimuli used here are inanimate objects. Previous data on left/right bias in relation to social and nonsocial stimuli are in line with current findings (see Dharmaretnam and Andrew, 1994) but then the argument of sociality cannot explain the sex differences since females have left-eye bias to an inanimate object. Current findings sometimes are in agreement with some but not all the previous literature.– In relation to the previous point, in the series of experiments presented here, authors did not test different types of stimuli (social vs nonsocial). This experiment could be relevant to support authors' argument of females being more interested to social stimuli.

In line with the Reviewer’s point that given its adaptive value, this ability should be also present in males, we do not believe that males cannot represent absence, we believe that in this specific experiment we did not manage to tackle on this ability in males, as it may be masked by a decreased attention to familiar social stimuli (Vallortigara, 1992; Versace, Spierings, Caffini, ten Cate and Vallortigara, 2017). Now we emphasize this issue on page 13:

“It is important to note that we do not have theoretical reasons to assume that males cannot represent absence, instead, we believe that in this specific experiment, males’ abilities were masked by a decreased attention to familiar social stimuli.”

Similarly to earlier studies, here we interpret the left-eye bias as associated to the detection of a novel object. Importantly in our case the test outcome could be encoded as new, only if the imprinting object was previously encoded as absent from that location. The finding that only females showed a left-eye bias in our study may be explained by the fact that they are more interested in familiar social stimuli than males, in the sense that they have likely paid more attention to, and encoded better the imprinting object that left the scene (that is, its absence). Males in contrast might have paid less attention to the imprinting object and its absence/presence, and therefore it is unclear whether they could encode the test outcome object as a new one.

Regarding the observation that we did not compare different types of stimuli (social vs; nonsocial) the point is well taken: we now acknowledge this in the manuscript. Indeed, it is an interesting question whether chicks would be equally sensitive to the absence of social and non-social; animate/inanimate objects. The stimuli we have used here were not just any objects, but imprinting objects with a social connotation, and were not static objects (they were moved by the experimenter hanging on an invisible string). While we did not test directly how exactly chicks in our experiment encoded the unexpectedly appearing objects (animate or inanimate), we agree that comparing different types of stimuli (social and nonsocial) is a great topic for future studies.

We added the following information to the manuscript on page 13:

“A further emerging question is how exactly this unexpectedly emerging object was encoded and whether similar effects would emerge for encoding the absence of nonsocial stimuli. While the present studies were not designed to directly test whether unexpectedly emerging object was perceived as an animate social individual or an inanimate object, further studies might focus on this question by comparing different types of stimuli (social and nonsocial) and also investigate whether males and females would react differently to such items.”

Other commentsIn general, I found the introduction well written but since the behavioral analysis has been mainly set around the left-right eye use, the rationale of the experiment has not been sufficiently elaborated in the introduction. No predictions have been put forward to justify the statistical approach adopted by the authors. This leaves the reader a bit confused while reading the result section.I recommend authors to shorten the general introduction and to develop it more around the laterality issue and its relevance for the current study. Predictions should be better spelled out.

In line with this comment we now streamlined the introduction and spelled out the predictions on page 4-6:

“Regarding our first measurement, based on former research with human infants (Baillargeon, Spelke and Wasserman, 1985) and a study involving adult rooks (Bird and Emery, 2010), we expected longer looking for unexpected outcomes. For instance, Bird and Emery (2010) have found that rooks looked longer to unexpected events that violate the laws of physics (e.g. objects remaining in the air without any support) compared to expected events (e.g. objects in a support relation with other objects), indicating a violation of their expectation or surprise, a method commonly used in infancy research to study a wide range of competencies.

For our second measurement, we coded which eye the chicks used to inspect the expected and unexpected outcomes. Earlier research suggests that eye usage is modulated by the novelty of the object attended, and also by sex (Rogers and Anson, 1979; Dharmaretnam and Andrew, 1994; Vallortigara and Andrew 1991). A preferential use of the left eye (mainly feeding the right brain structures) is associated with response to novelty in birds with laterally-placed eyes such as domestic chicks (Rogers, Vallortigara and Andrew, 2013). Note that the selective involvement of structures in the right hemisphere when attending to novel stimuli is widely documented among vertebrates (review in Rogers et al., 2013), being likely a general feature inherited by early chordates (MacNeilage et al., 2009). In animals with laterally placed eyes and lack of callosum, such as birds, fish, reptiles and amphibians the brain asymmetry can be easily documented without any invasive procedure by simply measuring preferences in eye use (Vallortigara, 2000; Vallortigara and Versace, 2017; Vallortigara and Rogers, 2020). In the present study we expected a left-eye bias to the novel unexpected outcomes compared to the expected ones.

In addition, interestingly, lateralized sex differences have been repeatedly observed in response to novel objects. For instance, Vallortigara and Andrew (1991) documented stronger left-eye mediated choices of unfamiliar stimuli for males compared to females, and different preferences for unfamiliar and familiar objects between sexes. Lateralized sex differences have been found also by Dharmaretnam and Andrew (1994), where unfamiliar stimuli evoked left-eye bias in females, but not in males. Vallortigara and Andrew (1991) observed that lefteye (and binocular) males preferred unfamiliar objects, while left-eye (and binocular) females preferred familiar objects, whereas both males and females tested with the right eye did not exhibit significant preferences for familiar or novel stimuli. Hence, a potential modulation of sex in eye use must be considered for the exploration of unexpected vs expected scenes in our study as well.”

– The left-eye preference without physical object is present in females but not males, and it has been interpreted within the framework of female's more efficient capacity of social discrimination. There are few logical passages which require to be better explained.The first is that the imprinted object used in the current experiments is not a social stimulus. Therefore, one should test whether this left-eye bias is stronger for social vs nonsocial objects. Secondly, if computing absence is not domain specific (i.e. social), but a general capacity of the species, why males should not have it?Furthermore, the work by Darmaretnam and Andrew shows that females are more inclined to use the right eye for a conspecific which contradicts previous interpretation (on sex differences). In brief, the first attempt to interpret data is aimed at justifying why females have the left-eye bias (p.13), but then the same argument does not hold because the right-eye bias is more prominent for conspecifics and not for objects. I think authors should be coherent and avoid twisting the arguments because this part of the discussion is rather confused.

We thank the reviewer for this comment, which allows us to clarify the points of misunderstanding, which may stem from the fact that we are talking about two different representations of chicks at two different time-points. Specifically, in the critical condition, chicks first see the attachment object leaving the scene and then a ‘new’ object is unexpectedly revealed in the scene. The first, Representation 1 (R1) thus is about the familiar attachment object and its absence, and we argue it may be encoded similarly to representations of social objects. The second, Representation 2 (R2) is about encoding the ‘novel’ object, which given that is novel may be social or nonsocial. In particular, in the critical conditions of the experiments targeting absence representations (Exp 2-3-4) chicks first can encode the familiar attachment object as present in the scene – R1, which then leaves the scene (and possibly encode its absence). Afterwards, unexpectedly an identically looking object appears in the arena -R2, which we argue it must be encoded as a novel object (if and only if the imprinting object (R1) was encoded as absent, and given that we do not observe a left-eye bias effect when the imprinting object appears expectedly). Thus, the left-eye bias is measured to this test outcome object (R2) that is likely encoded as novel, and it may be nonsocial (congruent with Dharmaretnam and Andrew, 1994 finding a left-eye bias for nonsocial objects).

We have suggested that the finding that only females show left eye bias may be explained by the fact that they are more interested in familiar social stimuli than males, in the sense that they have likely paid more attention to and encoded better the imprinting object (R1) that left the scene (that is, its absence), and therefore encoded the test outcome object (R2) as novel, resulting in a left-eye bias. As the reviewer point out, earlier studies have found a left-eye bias for novel objects (Rogers and Anson, 1979; Dharmaretnam and Andrew, 1994) but a right eye bias for observing conspecifics in female chicks (Dharmaretnam and Andrew, 1994). Thus, a possible way to reconcile earlier and present findings would be to argue that in our case the new object was encoded as a nonsocial object.

Males in contrast have shown no eye bias, which might have been due to paying less attention to the imprinting object R1 and its absence/presence (given a decreased attention to familiar social partners Vallortigara, 1992; Vallortigara and Andrew, 1994), and therefore it is unclear whether they encoded the test outcome object as a new one. In line with the Reviewer’s last suggestion, we do not believe that males cannot represent absence, we believe that in this specific experiment we did not manage to tackle on this ability in males, as it may be masked by their decreased attention to familiar social stimuli in the encoding phase.

– One striking difference with humans, is that representing absence of objects emerge between one to two years. In chicks, this is present at 8 days of age. I am not sure it makes sense to compare developmentally chicks and humans. Such comparison could be extremely useful to understand whether different behavioral responses to violations are based on the same (homologous or analogous) mechanisms, or whether in chicks perceptual and cognitive processes are specializations evolved as specific adaptations to their environment. Overall, honey-bees has the concept of "empty sets" despite their nervous system is substantially different from that of other vertebrates. I think that it would be useful to have some conceptualizations of how the behavioral phenomenon described here in chicks is based on mechanisms that could have evolved independently in other species in order to accomplish simple cognitive processes useful in specific domains (e.g. foraging, tracking conspecifics). The linguistic concept of "zero" could be an extension of such basic components.

Here the Reviewer argues, if we understand well, that it may not be reasonable to compare developmentally chicks and humans**,** and that the linguistic concept of zero could be an extension of more basic components shared across species. We cannot agree more, we discuss human adult and developmental data to give the reader a clear picture about what further concepts may be built on such basic abilities, as well as about the studies that have targeted a similar phenomenon in at a preverbal stage in humans (though unsuccessfully till now). Indeed, what we aimed to propose was that there may be some basic processes that allow for forming spontaneous absence representations in various species, which in humans will also support the development of more abstract concepts. And we have provided some initial evidence for this in young chicks.

We added a clarifying sentence to our introduction at page 1:

“In this work we focus on the representation of absence that should, however, rely on a more basic capacity and possibly be part of the initial cognitive repertoire of different species and we will target such abilities in 8-day-old domestic chicks.”

– At the end of the abstract it is suggested that chicks form "thoughts" about the absence. I think this term could raise some doubts among scholars (not only linguists), and I think its use here is neither necessary nor appropriate.

We have changed the term thoughts to representations.

– Figure 2B and Figure 4 probably contain asterisks to mark differences. They are almost invisible to the reader. If that is the case these figures should be remade and the legend should explain what the asterisks indicate.

We thank for noting this, the dots are the outliers, which is now specified in the figure legend.

– Along the text (Figures 1 and 3) authors used the terms 'encoding' and 'computing'. Are these terms, according to authors, indicating different concepts? If yes, these should be explained in the text, otherwise please be consistent with one of them.

To clarify the difference we changed the name of Experient 3 from “Computing absence” to “Perceptual Comparison” and modified the manuscript on page 8 the following way:

“According to one possibility, chicks in Experiment 2 (Encoding Absence) might have not formed an actual representation that there was no object behind the screen, instead their reaction to the unexpected appearance could have been perceptually driven and derived from detecting the mismatch between the memory of the empty space behind the screen (i.e., an iconic, picture-like representation of the empty floor and the wall of the testing arena) and the perceived outcome (i.e., the scene with the object). This possibility, however, would predict no left eye bias in a situation where there was no possibility for perceptually encoding the empty space. In contrast, left-eye bias without relying on the percept of an empty space would point to a more complex mental representation of absence that is inferred from the sequence of events (object going in/going out) (Experiment 3. Absence vs. Perceptual Comparison).”

– I found the last third of the discussion (from: "While our data provide……" until the end) rather speculative. It should be shortened into one paragraph focusing on the possibility of a categorical encoding of absence in the species.

We consider this part important and possibly stimulating for future research, therefore, we have decided to keep it, while we have tried to streamline it and acknowledge that is speculative. We modified the manuscript on page 14 the following way:

“However, we consider important briefly discussing the theoretical possibilities to embed our work in the literature and to stimulate further research. Following Nieder’s taxonomy (Nieder, 2016), the chicks in our study might have encoded absence either in a numerical or a categorical representational format. Regarding the numerical format, the approximate number system provides numerical representations to pre- and non-linguistic creatures as well (Dehaene, 1997; Haun et al., 2011; Brannon and Merritt, 2011; Brannon and Roitman, 2003). Absence could be represented by this system as an approximate numerical value of less than one. The other possible way to capture absence is via categorical representations, which are likely recruited in perceptual decision tasks to contrast the absence of an object to its presence (Merten and Nieder, 2012). However, the exact coding mechanism underlying categorical representation of absence and presence was not previously addressed. Whether specific pairs of contrary concepts (i.e. presence/absence) or general category-forming processes support such abilities is the potential target of future research.”

Reviewer #2 (Recommendations for the authors):Before publication, I suggest the authors clarify answers to the following specific queries:IntroductionOn the first page, the authors state that "by the age of 5 children seem to successfully operate with counterintuitive concepts like zero and nothing". The authors need to specify why this is counterintuitive, or remove such a statement (especially when on page 2 they say, "Things can be present or absent, yet how these intuitively simple opposing categories are formed and encoded is largely unexplored"). Maybe introducing the idea of contrary concepts here, instead of leaving that for the Discussion, could be helpful?

Here the Reviewer points out that we cannot say that a concept can be counterintuitive and simple at the same time. Indeed, we take the point, and we now clarify that more complex representations such as zero or nothing, which is not trivial how the young learners may arrive to, may build on more simple representations of presence and absence (page 1: “In this work we focus on the representation of absence that should, however, rely on a more basic capacity and possibly be part of the initial cognitive repertoire of different species and we will target such abilities in 8-day-old domestic chicks.”), also introduce contrary concepts in the introduction (page 2-3:“Bermúdez (2003) proposed that contrary concepts, like absence/presence might be available even for non-human animals. Such contrary concepts encompass alternatives that are mutually exclusive (e.g. nothing can be present and absent at the same time) and may support specific inferences. The availability of contrary concepts in pre- and non-linguistic animals has not been targeted by researchers, nevertheless, extensive research cumulating for over a century suggests that various species are able to exploit the presence and absence of stimulus (Pearce, 2011). However, clear evidence that absence is explicitly represented is scarce.”)

Previous research with rooks is mentioned briefly in one sentence near the end of the Introduction. But if this has already been done in another bird species, how does doing this in chicks further significantly advance knowledge? Or, was the rook research with adult birds rather than neonates?

Following the suggestion, we now describe in more detail the study with rooks (Bird and Emery, 2010), which is, to our knowledge, the only study that has used looking times in avian species to investigate a completely different question, specifically rooks’ understanding of physical support relations. We modified the manuscript on page 4-5:

“Regarding our first measurement, based on former research with human infants (Baillargeon, Spelke and Wasserman, 1985) and a study involving adult rooks (Bird and Emery, 2010), we expected longer looking for unexpected outcomes. For instance, Bird and Emery (2010) have found that rooks looked longer to unexpected events that violate the laws of physics (e.g. objects remaining in the air without any support) compared to expected events (e.g. objects in a support relation with other objects), indicating a violation of their expectation or surprise, a method commonly used in infancy research to study a wide range of competencies.”

A little more background review on eye usage with respect to novelty would be useful. What is the theory behind why this occurs? Is it specific to birds, and if so, why?

We have now included a paragraph on this issue. In addition, we also discuss in more detail lateralization studies in chicks on page 5-6:

“For our second measurement, we coded which eye the chicks used to inspect the expected and unexpected outcomes. Earlier research suggests that eye usage is modulated by the novelty of the object attended, and also by sex (Rogers and Anson, 1979; Dharmaretnam and Andrew, 1994; Vallortigara and Andrew 1991). A preferential use of the left eye (mainly feeding the right brain structures) is associated with response to novelty in birds with laterally-placed eyes such as domestic chicks (Rogers, Vallortigara and Andrew, 2013). Note that the selective involvement of structures in the right hemisphere when attending to novel stimuli is widely documented among vertebrates (review in Rogers et al., 2013), being likely a general feature inherited by early chordates (MacNeilage et al., 2009). In animals with laterally placed eyes and lack of callosum, such as birds, fish, reptiles and amphibians the brain asymmetry can be easily documented without any invasive procedure by simply measuring preferences in eye use (Vallortigara, 2000; Vallortigara and Versace, 2017; Vallortigara and Rogers, 2020). In the present study we expected a left-eye bias to the novel unexpected outcomes compared to the expected ones.

In addition, interestingly, lateralized sex differences have been repeatedly observed in response to novel objects. For instance, Vallortigara and Andrew (1991) documented stronger left-eye mediated choices of unfamiliar stimuli for males compared to females, and different preferences for unfamiliar and familiar objects between sexes. Lateralized sex differences have been found also by Dharmaretnam and Andrew (1994), where unfamiliar stimuli evoked left-eye bias in females, but not in males. Vallortigara and Andrew (1991) observed that lefteye (and binocular) males preferred unfamiliar objects, while left-eye (and binocular) females preferred familiar objects, whereas both males and females tested with the right eye did not exhibit significant preferences for familiar or novel stimuli. Hence, a potential modulation of sex in eye use must be considered for the exploration of unexpected vs expected scenes in our study as well”.

MethodsWhy were 8-day-old chicks the age chosen? Is there precedent in previous research for the eye bias to be present starting then (as hinted at in the Discussion), is this the crucial time period after Imprinting/Familiarization for Test for some particular reason, etc?

As also mentioned in the reply to the Editor, the age we have used was determined by the paradigm we have applied, as it was the earliest age we could test chicks with this specific imprinting paradigm that involved a confining cylinder at test. Moreover, chicks of both sexes have strong lateralized responses at this age (Dharmaretnam and Andrew, 1994). We now clarify this on page 4 (“The age was determined by the specific paradigm we used in the present work (see the details of imprinting and familiarization with the apparatus in the Materials and methods) and the dependent measures we targeted (looking time and lateralization index). This is also the age when chicks were found to show strong lateralized responses to familiar and unfamiliar objects (Dharmaretnam and Andrew 1994)”). While this experimental setting seems more suitable for testing relatively older (8-day-old) chicks, we do not have theoretical reasons to expect developmental differences between newly hatched and 8-day-old chicks’ capacity to represent absence of objects. However, it would be interesting to follow up the present work and develop a paradigm suitable for testing this question in newly hatched chicks.

ResultsIt is interesting sex was a factor in the ANOVAs, and did sometimes have an effect. Was this predicted? It is great that later in the Discussion the authors speculate why there may be sex-dependent representation of absence in chicks, but I think there needs to be background information included earlier so the reader can understand why this was part of the Analysis plan.

Following this suggestion, we have now included a discussion on sex differences in chicks’ lateralization index in earlier studies targeting other questions in the introduction, as well as clarified our predictions. While we have no theoretical reasons to expect sex differences in the ability to represent absence, given that sex differences are rather common in earlier lateralization studies involving familiar and unfamiliar stimuli, we have included it in the analysis. We added the following modification to the manuscript on page 5-6:

“In addition, interestingly, lateralized sex differences have been repeatedly observed in response to novel objects. For instance, Vallortigara and Andrew (1991) documented stronger left-eye mediated choices of unfamiliar stimuli for males compared to females, and different preferences for unfamiliar and familiar objects between sexes. Lateralized sex differences have been found also by Dharmaretnam and Andrew (1994), where unfamiliar stimuli evoked left-eye bias in females, but not in males. Vallortigara and Andrew (1991) observed that left-eye (and binocular) males preferred unfamiliar objects, while left-eye (and binocular) females preferred familiar objects, whereas both males and females tested with the right eye did not exhibit significant preferences for familiar or novel stimuli. Hence, a potential modulation of sex in eye use must be considered for the exploration of unexpected vs expected scenes in our study as well.”

I am also just generally confused about this speculation. Is the imprinting object not a social/conspecific equivalent object? How then would we really predict (and interpret!), based on previous literature, whether to expect a left or right side bias? Is it perhaps a weighted prediction (e.g., the weight of importance for predicting absence of objects in females is more important than the weight of right-eye-bias predictions…but for males, these weights may be more evenly distributed between sides?) This needs to be clarified.

We have now better clarified the issue, i.e. the two different representations we assume that chicks formed in this task if they represented absence, that we also mention in response to Reviewer 1. The points for a possible misunderstanding may stem from these two different representations of chicks at two different time-points in the study. One of these, Representation 1 (R1) is about the attachment object, which we think it may be encoded similarly to representations of social objects. This is based on the fact that chicks treat artificial imprinting objects as naturalistic imprinting objects, following them and producing contact calls when separated from them (Vallortigara, Versace, 2018). If this object was encoded as absent from the scene in the Unexpected appearance conditions of Experiments 2,3,4, the object appearing in the test outcome phase must be encoded as a ‘novel’ object and represented differently (Representation 2; R2). Given that this object is novel, it may be nonsocial, or to the minimum less social than R1 (on which chicks rely in the Expected appearance condition, i.e. when they represent the imprinting object being behind the occluder and they see it there in the outcome).

Regarding the effects of sex, we have suggested that the finding that only females show left eye bias may be explained by the fact that they are more interested in familiar social stimuli than males (Vallortigara, 1992; Versace, Spierings, Caffini, ten Cate and Vallortigara, 2017; Vallortigara and Andrew, 1994), in the sense that they have likely paid more attention to and encoded better the imprinting object (R1) that left the scene (that is, its absence), and therefore encoded the test outcome object (R2) as novel, resulting in a left-eye bias. In contrast, males have shown no eye bias, which might have been due to paying less attention to the imprinting object R1 and its absence/presence (given a decreased attention to familiar social partners), and therefore it is unclear whether they encoded the test outcome object as a new one. We added the following modifications to the manuscript on page 12-13:

“While we expected dominant left eye usage in response to unexpected events in general, this measurement was sensitive specifically to violations of expectations of absence.

We propose that the left eye bias reflects chicks’ attempt to identify a novel object that appeared at a location where ‘nothing’ was expected to be. Two arguments support this interpretation. First, while our initial prediction was left-eye bias in response to unexpected events in general, this behavior was found only for outcomes involving an object (Experiment, 2, 3 and 4), but not when the chicks were confronted with an empty space, where an object should have been present (Experiment 1). Previous studies reporting left-eye bias in domestic chicks (Rogers and Anson, 1979; Dharmaretnam and Andrew, 1994) also involved objects as stimuli. According to our knowledge, no former work found left-eye preference in domestic chicks without a physical object. This suggests that this specific behavior is closely connected to the perceptual processing of objects, which in our case were most likely social (imprinting) objects. Second, the left eye bias was prominent in females compared to males, and such sexdependent differences are congruent with findings in social discrimination experiments with chicks (Vallortigara, 2017; Vallortigara and Andrew, 1994; Versace, Spierings, Caffini, ten Cate and Vallortigara, 2017; Pallante, Rucco and Versace, 2020). It is important to note that we do not have theoretical reasons to assume that males cannot represent absence, instead, we believe that in this specific experiment, males’ abilities could have been masked by a decreased attention to familiar social stimuli in the encoding phase, or by other factors. Indeed, female chicks are more interested in familiar social partners, while males are more interested in unfamiliar ones (Vallortigara, 1992), and these preferences are also observed in experiments using artificial social partners (Vallortigara and Andrew, 1994) and different sensory modalities (Versace, Spierings, Caffini, ten Cate and Vallortigara, 2017).”

In Study 2, why are there no untransformed mean looking time results given in the text, as there were for Study 1?

Study 2 was designed to investigate the possible boundary conditions of chicks’ absence representations as reflected by the left eye bias observed in Experiment 2-Study 1, and asked further questions about the underlying processes. Since based on Study 1 we did not expect any effects in the overall looking time measurements, we did not target this in Experiments 3 and 4. In hindsight, however, we agree with the reviewers that the readers may benefit from this information as well. Thus, for completeness, we now report the means and the standard error values of the looking time data (merged and separately for males and females) for all experiments in the supplementary information. However, we would like to note that our conclusions are based on the lateralization index measurement that is independent of the overall looking time. We added the following modification to the manuscript on page 11:

“As Study 2 was specifically designed to investigate questions related to the nature of absence representations as reflected by the left eye bias observed in Experiment 2, in Experiment 3 and 4 we did not aim to target overall looking times and neither we had any predictions regarding these. However, for completeness, we report these in the Supplementary Material”.

DiscussionOn page 12, the authors should clarify whether "Thus, remarkably, we found sex-dependent evidence for young chicks' ability to represent the absence of an object in their eye usage" applies to all Studies, or just a subset.Overall, this is subject matter interesting to readers of eLife. I think this could be suitable for publication if these concerns were addressed well.

In the discussion we refer to all the three studies that have targeted absence representations (Experiment 2,3,4), which we now clarify on page 12 (“Thus, remarkably, we found sexdependent evidence for young chicks’ ability to represent the absence of an object in their eye usage in the three experiments targeting absence representation (Experiment 2, 3, 4)”).

We would like to thank again for the insightful comments, which have improved the paper.

Reviewer #3 (Recommendations for the authors):The experiments are well conducted and discussed, thus I have very limited suggestions.– I suggest the Authors clarify which are the cognitive mechanisms underlying the representation of the absence of objects. In the initial part of the introduction, the approximate number system and the object tracking system are presented as the cognitive systems underlying numerical comprehension in animals. Nevertheless, it is unclear which system was considered responsible for the representation of absence. Moreover, I suggest explaining why "chicks' representations of absence seem to involve complex mental computations and possibly complex representations, that go beyond the object tracking system". This system is devoted to object representation, thus it seems the most suitable to be triggered by the presentation of an imprinting object. This system can estimate the number of perceived objects (also in this animal model), possibly starting from an empty set, thus it is unclear why it cannot be responsible for a representation of the absence of objects. Considering the introduction opens with an illustration of the cognitive systems underlying numerical cognition, I suggest integrating these findings with the previous literature showing numerical competences and the respective cognitive mechanisms in the animal model used in this study. Moreover, since this study is based on the prerequisite of object permanence, it would be interesting to discuss this evidence in more detail.

Following the Reviewer’s suggestion we now discuss in more detail the cognitive systems responsible for absence representations in the introduction as well, and explain why the object tracking system cannot deal with such representations. In particular, we clarify that the descriptions of the object tracking system, that we are aware of, only allows for tracking the presence and the whereabouts of an object (e.g. an object indexed at a particular spatiotemporal location; Kahneman, Treisman and Gibbs, 1992; Scholl and Pylyshyn, 1999), but do not encode the absence of a particular object, therefore representations of empty sets are beyond the scope of such systems (see also Exp 4 in the current study). Thus, our discussion is not focused on the system responsible for object permanence but on other possible cognitive systems. We added the following modification to the manuscript on page 3:

“While in these tasks the representation of an object being present can be supported by the object tracking system (Kahneman, Treisman and Gibbs, 1992), it is unclear how a specific object that is absent could be encoded by the same system (note that simply discarding the object file results in no representation whatsoever and it is not equivalent to representing, for instance, ‘the lion is absent’). In fact, the representation of absence might be beyond the scope of this system, as it operates with spatio-temporal information of the items, which absent objects do not have”.

– I also suggest to better distinguish the concept of zero and the representation of absence of objects in the introduction.

We now specify further that the concept of zero and absence representations may be very different, however the former more abstract representations may rely on the latter more basic ones (on page 1 “In this work we focus on the representation of absence that should, however, rely on a more basic capacity and possibly be part of the initial cognitive repertoire of different species and we will target such abilities in 8-day-old domestic chicks”).

– Scientific evidence on numerical competences in domestic chicks is mainly provided in the very first 5 days of life. Thus, I suggest specifying why here, chicks were tested at 8 days of life. Is this choice related to the employed paradigm or to an assumed underling cognitive development necessary for the chicks to grasp this concept?

The age we have used was determined by the paradigm we have applied, as it was the earliest age we could test chicks with this specific imprinting paradigm that involved a confining cylinder at test. Moreover, chicks of both sexes have strong lateralized responses at this age (Dharmaretnam and Andrew, 1994). We now clarify this on page 4 (“The age was determined by the specific paradigm we used in the present work (see the details of imprinting and familiarization with the apparatus in the Materials and methods) and the dependent measures we targeted (looking time and lateralization index). This is also the age when chicks were found to show strong lateralized responses to familiar and unfamiliar objects (Dharmaretnam and Andrew 1994)”).

– It is unclear to me why Experiment 1 is labeled as "encoding presence" and Experiment 2 "encoding absence".– The results of the comparison between the Expected disappearance (Exp 1) and unexpected appearance (Exp 2) might merit a deeper discussion. They share the initial part of the presentation of the stimulus, but differ for the outcomes (the presence or the absence of the object once the panel was lowered).

We specify (page 6) at the beginning of the description of Study 1 that in the critical condition of Experiment 1 chicks had to encode the presence of the imprinting object (as it went behind the occluder), while in the critical condition of Experiment 2 that had to encode its absence (as it left the scene) and we had specific predictions regarding how they would react to outcomes that are incongruent with such representations. The other two control conditions featured expected outcomes to which we have compared our critical conditions.

– I suggest reporting all the effect sizes and to modulate the discussion on the basis of their values.

We have added the missing effect sizes. All but one effect sizes of the reported ANOVAs are in the medium range. In Study 2 the interaction between Sex and Outcome has a small effect size, however, we do not draw strong theoretical conclusions in our Discussion based on this effect.

– Specify how the sample size was calculated.

Sample size was estimated based on a previous study with infants that has implemented a violation of expectation paradigm to measure absence representations via looking times (Wynn and Chiang, 1998), which we have used as inspiration for the present study.

– Italicize all species names in the references.

Thank you for all the constructive comments and suggestions, we have done that.